# Ciliary neurotrophic factor-mediated neuro-protection involves enhanced glycolysis and anabolism in degenerating mouse retinas

Kun Do Rhee[1,5], Yanjie Wang[1], Johanna ten Hoeve[2], Linsey Stiles [2], Thao Thi Thu Nguyen[1], Xiangmei Zhang[1], Laurent Vergnes[3], Karen Reue [3], Orian Shirihai [2], Dean Bok[1] & Xian-Jie Yang [1,4] ✉

Ciliary neurotrophic factor (CNTF) acts as a potent neuroprotective cytokine in multiple models of retinal degeneration. To understand mechanisms underlying its broad neuroprotective effects, we have investigated the influence of CNTF on metabolism in a mouse model of photoreceptor degeneration. CNTF treatment improves the morphology of photoreceptor mitochondria, but also leads to reduced oxygen consumption and suppressed respiratory chain activities. Molecular analyses show elevated glycolytic pathway gene transcripts and active enzymes. Metabolomics analyses detect significantly higher levels of ATP and the energy currency phosphocreatine, elevated glycolytic pathway metabolites, increased TCA cycle metabolites, lipid biosynthetic pathway intermediates, nucleotides, and amino acids. Moreover, CNTF treatment restores the key antioxidant glutathione to the wild type level. Therefore, CNTF significantly impacts the metabolic status of degenerating retinas by promoting aerobic glycolysis and augmenting anabolic activities. These findings reveal cellular mechanisms underlying enhanced neuronal viability and suggest potential therapies for treating retinal degeneration.

Ciliary neurotrophic factor (CNTF) has long been recognized as a potent neuroprotective agent in the vertebrate retina[1]. Enhancement of neuronal survival by CNTF has been demonstrated in multiple animal models of retinal degeneration, ranging from zebrafish to canine[2]. Interestingly, CNTF is effective in rescuing photoreceptor degeneration due to various underlying causes, including mutations in photoreceptor-specific genes and damages induced by strong light or neurotoxins[3–11]. In addition to enhancing the viability of photoreceptors, CNTF has been shown to increase the survival of retinal ganglion cells and promote retinal ganglion cell axonal regeneration in optic nerve crush or transection models[12–19]. Based on its significant and broad neuroprotective effects for retinal neurons,

an encapsulated cell implant producing a secreted form of recombinant human CNTF has been tested in clinical trials to treat hereditary and age-related retinal degenerative diseases[20–29]. The trials for retinitis pigmentosa (RP) with two CNTF doses for durations up to two years detected increased retinal thickness but did not observe efficacy using the best corrected visual acuity as the primary outcome[25]. However, recent reports of the CNTF trial for treating macular telangiectasia (MacTel) type 2 have described morphological and visual function improvements using multiple readout parameters[30,31]. The ongoing clinical trials also include treatment for glaucoma with retinal ganglion cell loss, a leading cause of blindness world-wide[32].

[1]Stein Eye Institute, Department of Ophthalmology, University of California, Davide Geffen School of Medicine, Los Angeles, CA 90095, USA. [2]Department of Molecular and Medical Pharmacology, University of California, Davide Geffen School of Medicine, Los Angeles, CA 90095, USA. [3]Department of Human Genetics, University of California, Davide Geffen School of Medicine, Los Angeles, CA 90095, USA. [4]Molecular Biology Institute, University of California, Los Angeles, CA 90095, USA. [5]Present address: Changpa Institute, Daegu University, Gyeongsan 38453, Korea. ✉e-mail: yang@jsei.ucla.edu

CNTF belongs to a subfamily of cytokines that share a tripartite receptor system, including the two transmembrane receptors gp130 and LIFRβ and a ligand-specific alpha receptor (CNTFRα)[33,34]. In developing and mature retinas, CNTF primarily stimulates the Jak-STAT and MEK-ERK signaling pathways[35–39]. Using cell type-specific gene ablation analysis in a mouse retinal degeneration model, we have shown that exogenous CNTF delivered to degenerating retinas initially activates STAT3 and ERK signaling in Muller glial cells through the gp130 receptor[40]. This initial event elicits an intercellular signaling cascade, which subsequently triggers gp130-mediated STAT3 activation in rod cells to promote photoreceptor survival[40]. Previous studies have also shown that the dosage and duration of CNTF treatment critically affect the rescued neurons, as a single dose injection of CNTF transiently affects the length of photoreceptor outer segments while prolonged exposure to high levels of CNTF can result in decreased visual function despite robust rescue of photoreceptors[7,41–43]. Molecular analyses have revealed that CNTF-induced STAT3 phosphorylation, followed by its dimerization and nuclear entry, significantly influences the retinal transcriptome, resulting in rapid elevation of transcripts involved in innate immunity and growth factor signaling, as well as reduced expression of genes involved in phototransduction and maintenance of photoreceptor identities[44].

The mature retina is among the most metabolically active compartments in the central nervous system. Photoreceptors have high energy demands for regulating membrane potentials in response to visual cues[45,46]. In addition, photoreceptors require continuous lipid and protein biosynthesis to sustain lifelong outer segment renewal[47]. Both rod and cone photoreceptors primarily consume glucose, which is supplied by the choroidal vessels through the retinal pigment epithelium (RPE)[48–50]. Fatty acids can also be a fuel source available to photoreceptors through the blood supply and the catabolic process of outer segment degradation by the RPE[51–53]. Accumulating evidence indicates that cone photoreceptor survival relies on glucose availability, which is partially dependent on neighboring rod cells[54,55]. Although mature photoreceptors contain a multitude of mitochondria distributed in the inner segments and at the synaptic termini, photoreceptors rely heavily on aerobic glycolysis under normal conditions[56–59]. Perturbing the glycolytic pathway regulatory enzymes phosphofructokinase (PFK) or lactate dehydrogenase (LDH) can result in deficits in outer segment renewal[58]. Ablation of genes encoding the glycolytic pathway enzymes hexokinase (HK2) in rod cells[60] or pyruvate kinase (PKM2) in rod or cone cells[61,62] leads to photoreceptor dysfunction and degeneration in aging mice. Furthermore, the protection of photoreceptors from strong light-induced damage involves AMPK activity, which is regulated by the cellular AMP to ATP ratio[63]. The cumulative data thus suggest that the cellular metabolic status of photoreceptors plays a crucial role in photoreceptor viability and function.

To elucidate cellular mechanisms underlying CNTF-dependent enhancement of neuronal viability, we have investigated the impact of CNTF signaling on retinal metabolism in a mouse model of retinitis pigmentosa, in which rod death precedes cone loss. By delivering the same secreted human CNTF used in clinical trials followed by molecular, cellular, and biochemical analyses, we demonstrate that CNTF treatment effectively influences the metabolic status of the degenerating retina, leading to elevated aerobic glycolysis and enhanced anabolism. These findings thus reveal a fundamental cellular mechanism by which a neurotrophic factor promotes neuronal viability in disease conditions.

## Results

### CNTF treatment alters the morphology of rod photoreceptor mitochondria

To study the influence of CNTF on retinal metabolism under photoreceptor degeneration conditions, we used a mouse model of retinitis pigmentosa that expressed the dominant mutant pheripherin2 transgene Prph2(P216L) in the wild type photoreceptors[64]. This transgenic mouse model exhibits shortened photoreceptor inner segments, rudimental outer segments, and a relatively slow loss of photoreceptors, referred to as "retinal degeneration slow" (rds) herein. After mice reach adulthood at postnatal day 25 (P25), the rds mutant retina contains approximately 80% of photoreceptors in number compared to the wild type. By P45, the loss of photoreceptors in the rds mutant has reached 50%. Subretinal delivery of a lentivirus expressing secreted CNTF (LV-CNTF) at P25 halts the degeneration process, resulting in the pan-retinal rescue of rod cells and partial restoration of the outer segment as well as the inner segment[40], where the majority of photoreceptor mitochondria reside.

To examine the influence of CNTF on rod mitochondria, we genetically labeled rod mitochondria by crossing Rho-iCre mice[65] with the PhAM mouse line that encodes a Cre-dependent mitochondrial targeting fluorescent reporter dendra2[66]. The resulting mouse retina showed specific labeling of rod mitochondria with the dendra2 reporter in the inner segments and the ribbon synapses of wild type rod photoreceptors (Fig. 1a, b). The rds mutant retinas exhibited similar rod mitochondrial labeling, but with more ectopically located mitochondria within the outer nuclear layer where rod photoreceptor soma resided (Fig. 1a). To characterize the morphology of rod mitochondria, we performed super-resolution structured illumination microscopy (SIM). The wild type rod mitochondria within the inner segment presented elongated cylindrical morphology, whereas those in the synaptic terminals were larger in size and granular in shape (Fig. 1c and Supplementary Videos 1, 2). In contrast, rod mitochondria in the rds mutant inner segment exhibited shortened and fragmental morphology (Fig. 1c and Supplementary Video 3). The LV-CNTF treatment of the rds retina resulted in elongation of the inner segments containing enlarged mitochondria both in the inner segments and in the synaptic termini (Fig. 1c and Supplementary Video 5), whereas control virus LV-IG treatment did not result in photoreceptor rescue or significant changes of mitochondrial morphology (Fig. 1c and Supplementary Video 4). TEM analysis also confirmed these morphological alterations detected by SIM (Fig. 1d).

Since mitochondria morphology may correlate with their functions[67], we characterized the morphological features of mitochondria in rod inner segments using acquired SIM imaging data (Fig. 2 and Source Data for Fig. 2). The MitoMap analysis[68] confirmed that the rds mutant mitochondria showed a marked departure from the wild type with increased sphericity and distribution isotropy, but decreased compactness and surface to volume ratio (Fig. 2a). Principal component analysis indicated that in CNTF-treated rds retinas, morphological features of inner segment mitochondria displayed more resemblance to the wild type (Fig. 2b). However, in CNTF-treated rds retinas the mitochondrial population overall still exhibited significant differences from the wild type with regard to their compactness, surface to volume ratio, and distribution isotropy (Fig. 2a).

### CNTF treatment affects mitochondrial respiration and suppresses respiratory complex activity

We next investigated the cellular respiration status using an Agilent XF96 Extracellular Flux Analyzer to simultaneously measure the oxygen consumption rate (OCR) and the extracellular acidification rate (ECAR), which may serve as a surrogate indicator for glycolysis (Fig. 3 and Source Data for Fig. 3). Compared to the wild type, the rds mutant retina showed decreased basal OCR and significantly increased basal ECAR as early as P17 before overt degeneration of photoreceptors (Fig. 3a). Both the OCR reduction and ECAR elevation persisted through P35 during the continuous loss of rod cells in the outer nuclear layer. Despite the lengthening of rod inner segments in the rds retina, CNTF treatment did not result in an increased OCR, but significantly

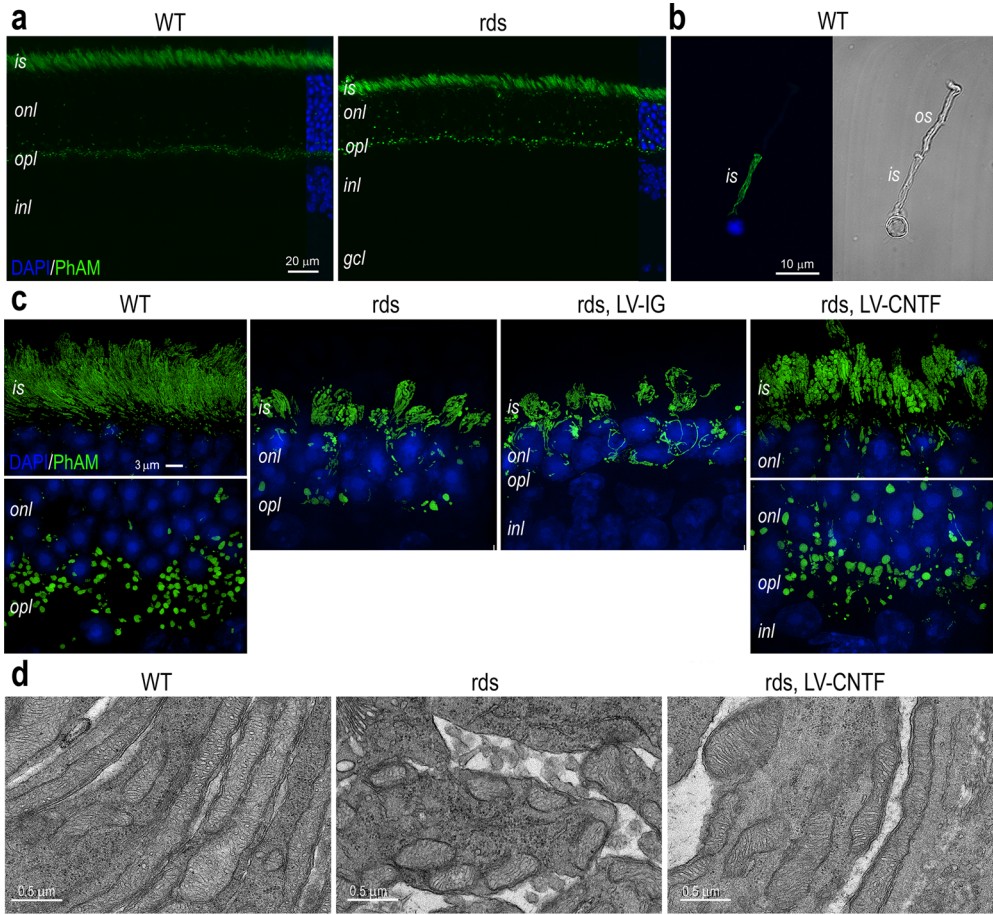

**Fig. 1 | CNTF-induced mitochondrial morphological changes in rod photoreceptors. a** Confocal microscopy images of P70 WT and rds retinas with the mitochondrial PhAM reporter activated in rod photoreceptors. **b** A dissociated P35 WT rod photoreceptor with PhAM reporter shown as Airyscan fluorescent image (left) and brightfield image (right). **c** SIM images show PhAM-labeled rod mitochondria in WT, rds, and rds mutant injected with LV-IG or LV-CNTF at P28 and harvested at P96. **d** TEM images of rod inner segments at P52 in WT, rds, and rds mutant injected with LV-CNTF. All image analyses used three independent retinas (N = 3). Scale bars: **a** 20 µm; **b** 10 µm; **c** 3 µm; **d** 0.5 µm. WT wild type, rds Prph2 P216L mutant, is inner segment, os outer segment, onl outer nuclear layer, opl outer plexiform layer, inl inner nuclear layer, gcl ganglion cell layer.

elevated ECAR (Fig. 3b), suggesting possible elevation of glycolysis. To examine whether CNTF affected oxidative phosphorylation, we first assayed the basal OCR, and then determined the amount of ATP-linked respiration and the maximal OCR by sequential additions of the ATP synthase inhibitor oligomycin followed by the proton gradient uncoupler FCCP (Fig. 3c). The OCR measurements indicated that CNTF treatment did not enhance the basal or maximal OCRs, but caused a reduction of ATP-linked OCR (Fig. 3c, d). These results suggested that CNTF treatment did not enhance mitochondrial respiratory chain-dependent oxygen consumption in the rds mutant retina.

To validate the retinal tissue assay results, we analyzed mitochondrial respiration using isolated retinal mitochondria in the presence of different respiratory chain substrates and inhibitors (Fig. 4 and Source Data for Fig. 4). In the presence of pyruvate and malate, substrates for complex I, mitochondria from rds retinas injected with the control virus LV-IG showed lower complex I-linked respiration compared to the wild type (Fig. 4a, b). LV-CNTF treatment of the rds retina caused a further reduction of complex I activity (Fig. 4a, b). When complex I was inhibited by rotenone and succinate was supplied, a slight reduction of complex II activity was detected in the rds retina (Fig. 4c, d). When complex I and complex III were both inhibited by rotenone and antimycin A to test complex IV activity, mitochondria from CNTF-treated rds retina showed a significant deficit when compared to the wild type (Fig. 4e, f). Together, these results demonstrated that the rds retina had an impaired mitochondrial respiratory chain, and

CNTF treatment did not improve but instead caused a further suppression of mitochondrial respiratory chain function in the rds mutant.

## CNTF elevates energy production, aerobic glycolysis, and anabolic metabolites

To obtain a comprehensive assessment of the retinal metabolic status, we performed metabolomics analysis for wild type, rds, and rds retinas treated with either the control LV-IG or LV-CNTF virus. Quantification of cellular metabolites with glucose as a fuel revealed a dramatic impact of CNTF signaling on retinal metabolism (Supplementary Fig. 1 and Source Data of Metabolomics). Compared to the wild type, the rds mutant retina had a lower level of ATP whereas CNTF-treated rds retinas contained 1.5-fold of the wild type level of ATP, and consequently a lower ADP/ATP ratio (Fig. 5a; Supplementary Fig. 2 and Source Data for Fig. 5). In addition, the energy currency phospho-creatine (p-creatine), which is normally present at high concentrations in the brain and muscles to act as an energy buffer to quickly regenerate ATP[69,70], was increased to 3.9-fold of the wild type level in CNTF-treated rds retinas (Fig. 5a). Furthermore, CNTF restored the level of GTP in the rds retina, which was reduced to 45% of the wild type level, and elevated 5′-methylthioadenosine to 1.4-fold of the wild type level (Fig. 5b and Supplementary Fig. 4). Consistent with the notion that CNTF signaling promoted aerobic glycolysis, we detected increases of glycolytic pathway intermediates fructose1,6 bisphosphate (F1,6BP), 3-phosphoglycerate (3PG) and phosphoenolpyruvic acid (PEP) (Fig. 5c

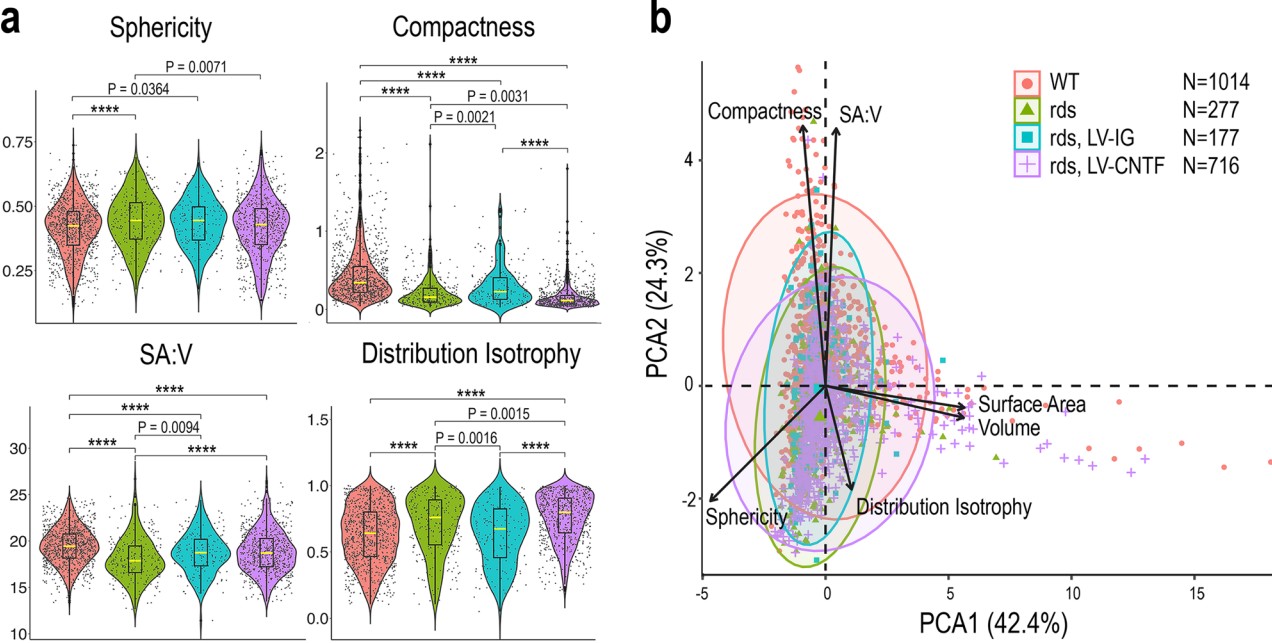

**Fig. 2 | Characterization of rod photoreceptor mitochondrial morphological features. a** Quantification of PhAM-labeled mitochondria in rod inner segments captured with SIM in WT, rds, and rds retinas ($N = 3$) treated with LV-IG or LV-CNTF. Violin plots show geometrical features of the individual mitochondrion, including sphericity, compactness, ratio of surface area to volume (SA:V), and distribution isotropy. Dots represent individual mitochondria data points. Boxes represent 50% of the mitochondria, and yellow lines represent median values. The horizontal bounds of the box indicate 25th and 75th percentiles. The lower whiskers comprise the minimum data value within 1.5 times the interquartile range below the 25th percentile. The upper whiskers include the maximum value of the data within 1.5 times the interquartile range above the 75th percentile. One-way ANOVA and Tukey all-pairs test were applied with adjusted $P$ values shown. $P < 0.0001$ is indicated as ****. **b** Principal component analysis (PCA) separates rod mitochondrial geometry. Separation in PCA1 is mainly driven by surface area, volume, and sphericity, while compactness and SA:V contribute to separation in PCA2. $N$ in **b** represents the number of mitochondria used for the analysis shown in this figure.

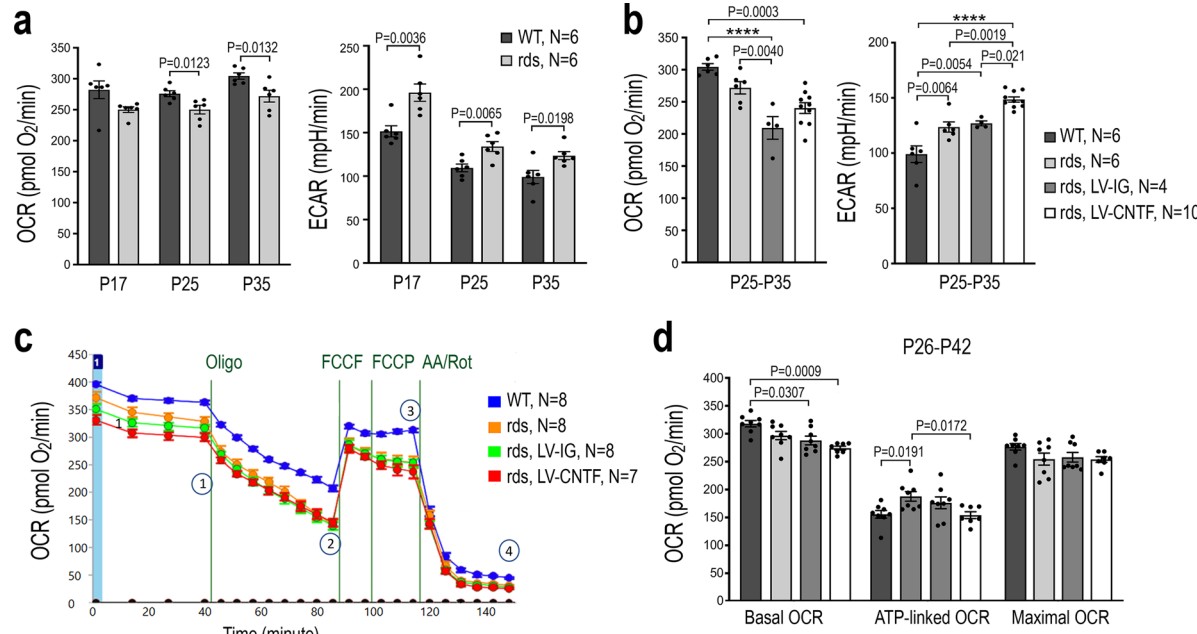

**Fig. 3 | Influences on retinal cell respiration by the rds mutation and CNTF.**
**a** Seahorse metabolic flux analysis measuring basal OCR (left) and ECAR (right) for WT and rds retinas at P17, P25, and P35. For each age and genotype, $N = 6$ independent retinas were used. $P$ values from the two-tailed student $t$-test are indicated. **b** Effects of CNTF on basal OCR and ECAR in rds retinas treated with LV-IG or LV-CNTF from P25-P35 compared to WT and non-treated rds retinas. Independent sample numbers $N$ and adjusted $P$ values from one-way ANOVA and Tukey all-pairs test are shown with $P < 0.0001$ indicated as ****. **c, d** Seahorse mitochondria stress tests to determine basal and maximum OCR, as well as ATP-linked OCR by sequential treatments with respiratory chain inhibitors oligomycin (oligo), uncoupling agent FCCP, antimycin A (AA), and rotenone (Rot) for WT, rds, and rds retinas treated with either LV-IG or LV-CNTF from P26-P42. **c** Seahorse assay tracings and the states indicating (1) basal OCR, (2) oligomycin inhibition of ATP synthase, (3) maximum OCR, and (4) total inhibition of respiration with AA and Rot. **d** Bar graphs show basal OCR, ATP-linked OCR, and maximal OCR. Independent sample numbers $N$ and adjusted $P$ values from two-way ANOVA and Tukey all-pairs test are shown. For **a**, **b**, **d**, data were presented as mean values ± SEM.

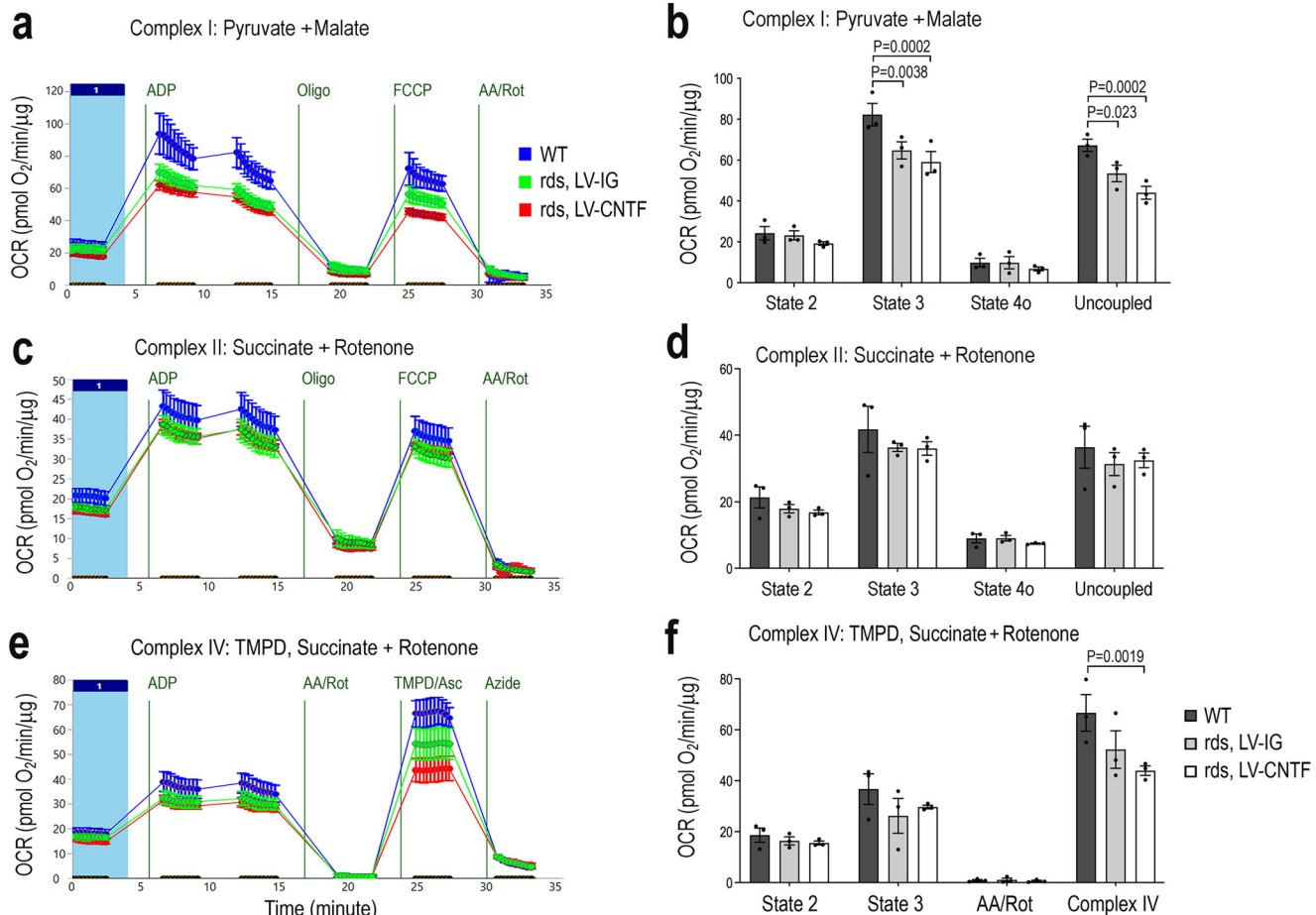

**Fig. 4 | Quantification of mitochondrial respiratory chain complex activities.** Mitochondria stress test using isolated mitochondria to examine the activity of each complex from WT and rds retinas treated with LV-IG or LV-CNTF from P25-P49. OCR (pmol/min/μg protein) were measured. **a**, **b** Complex I respirometry tracing and bar graph using pyruvate and malate as substrates. **c**, **d** Complex II-driven respiration measured in the presence of succinate and complex I inhibitor rotenone. **e**, **f** Complex IV state 3 respiration was measured in the presence of succinate and rotenone, following inhibition of complex I and III, and injection of TMPD/ascorbate to measure Complex IV activity. Independent samples $N = 3$, and each contains mitochondria from four retinas for all conditions. For **b**, **d**, **f**, data were presented as mean ± SEM. Adjusted P values derived from two-way ANOVA and Tukey all-pairs test are indicated.

and Supplementary Fig. 2). Furthermore, metabolomics analysis detected significant elevation of TCA cycle products, including aconitate, citrate, alpha-ketoglutarate, and malate (Fig. 5d and Supplementary Fig. 2). Importantly, CNTF treatment also altered the reduction-oxidation status by fully restoring the antioxidant glutathione (GSH) in the rds retina from 50% reduction to the wild type level (Fig. 5e and Supplementary Fig. 2).

Quantitative metabolomics analysis revealed that rds retinas treated with LV-CNTF, but not the control LV-IG virus, elevated cellular contents for the majority of amino acids to above the wild type levels (Fig. 5f and Supplementary Fig. 3). Similarly, multiple fatty acid biosynthetic intermediates that showed reduced levels in the rds retina were also elevated by CNTF treatment (Fig. 5g and Supplementary Fig. 2). The results of metabolomics analysis, therefore, indicated that CNTF signaling asserted a strong influence on retinal metabolism by promoting glycolysis, increasing energy supply, and enhancing anabolism in the rds mutant retina.

### CNTF signaling impacts the expression of metabolic genes and enzymatic activities

To evaluate whether CNTF signaling influenced metabolism at the transcription level, we performed high throughput RNA-sequencing and analyzed the retinal transcriptome of the rds mutant retinas treated with either the control LV-IG or LV-CNTF

from P25 to P35[44]. Compared to the wild type, LV-CNTF treatment led to increased expression of most glycolytic pathway transcripts, whereas LV-IG injection induced moderate gene expression elevation (Fig. 6a and Source Data for Fig. 6). CNTF treatment also elevated expression of many genes encoding the TCA cycle enzymes (Fig. 6b). Some of the genes involved in mitochondria stress and turnover, such as Pink1[71], was elevated in the rds mutant (Fig. 6c). CNTF treatment suppressed the expression of Pink1 in the rds mutant, and elevated expression of key mitochondrial transcription factor Tfam[72], suggesting that CNTF signaling influenced mitochondrial dynamics. Transcriptome analysis also revealed that most nuclear-encoded respiratory complex genes were up-regulated after CNTF treatment; however, several components for complex I, complex IV, and ATP synthase were downregulated compared to LV-IG treated rds retinas (Fig. 6d).

Next, we performed Western blot analysis to examine the Jak-STAT signaling and the status of enzymes in the glycolytic pathway. As expected, CNTF treatment resulted in increased phosphorylation of STAT3 at Y705 as well as elevated total STAT3 protein (Fig. 6e and Source Data for Fig. 6). Consistent with the observed CNTF-induced elevation of LDHa mRNA (Fig. 6a), the active form of LDHa with Y10 phosphorylation were increased to near two-fold of LV-IG treated rds retinas. These data thus further validated that CNTF signaling enhanced glycolysis in rds retinas.

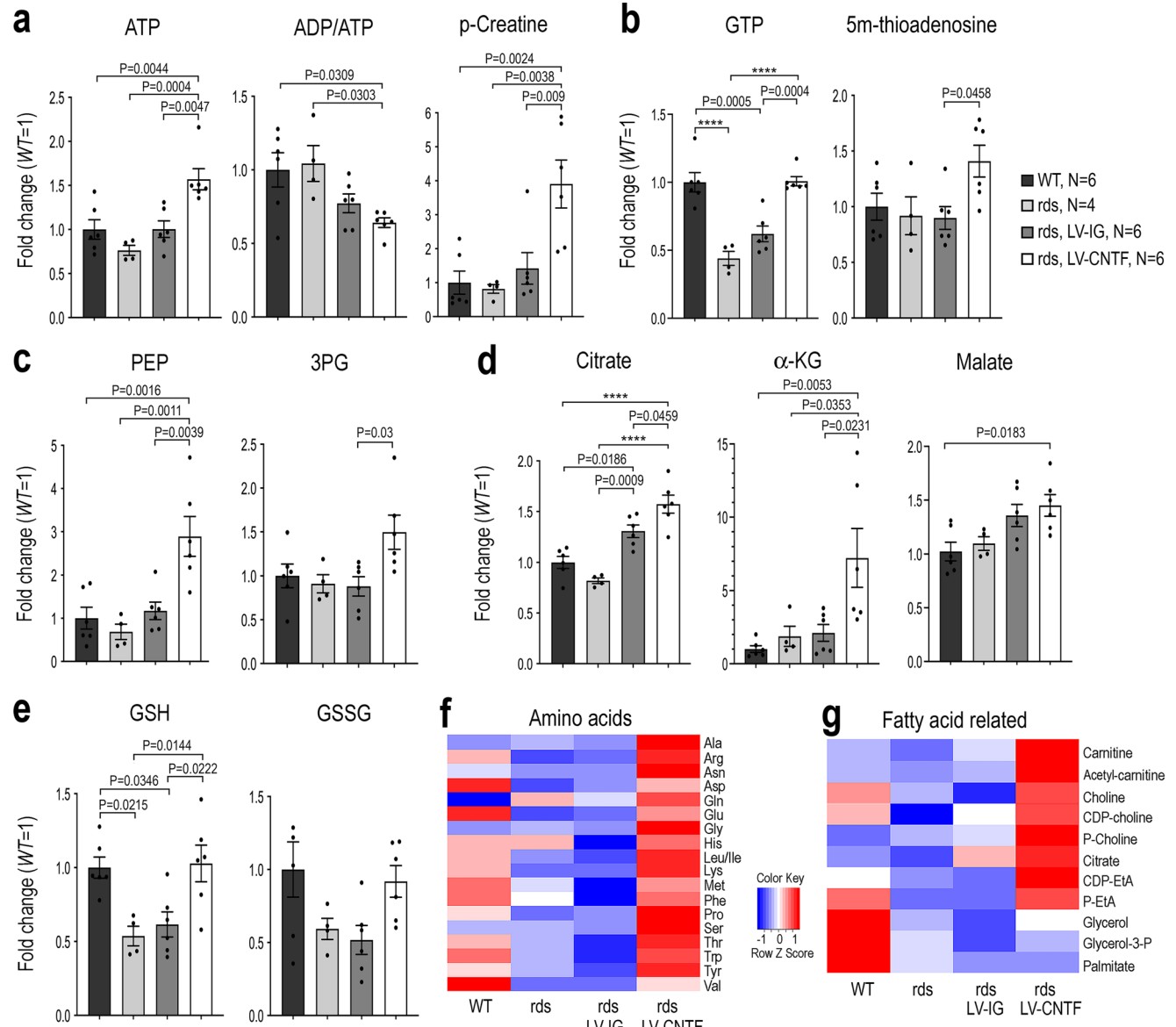

**Fig. 5 | CNTF-dependent enhancement of aerobic glycolysis and anabolism.** Metabolomics analysis of WT, rds, and rds mutant retinas treated with LV-IG or LV-CNTF from P25-P36 using glucose as a nutrient. **a** Energy metabolites ATP, phosphocreatine (p-creatine), and ADP/ATP ratios. **b** Nucleotide derivatives. **c** Glycolytic pathway intermediates 3-phosphoglycerate (3PG) and 2-phosphoenol pyruvate (PEP). **d** TCA cycle metabolites citrate, alpha-ketoglutarate (α-KG), and malate. **e** Redox-related metabolites glutathione (GSH) and glutathione bisulfide (GSSG). **f** Heatmap shows retinal amino acid contents. **g** Heatmap shows fatty acid metabolic intermediates. For **a**–**e**, data were presented as mean ± SEM. Independent retinal sample numbers (*N*) and adjusted *P* values based on one-way ANOVA and Tukey all-pairs test are shown with *P* < 0.0001 indicated as ****.

## Contribution of glycolysis and OXPHOS in wild type and degenerating retinas

Since the metabolic changes in degenerative retinas were not well characterized, we examined the metabolic contribution from aerobic glycolysis and mitochondrial respiration in the wild type and rds retinas. Metabolomics analyses were performed under conditions with various metabolic inhibitors (Fig. 7a and Source Data for Fig. 7). In the presence of the complex IV inhibitor sodium azide (Supplementary Fig. 5), amino acid contents in both the wild type and CNTF-treated rds mutant retinas were significantly reduced (Fig. 7b and Supplementary Fig. 6), indicating that the respiratory chain activities impacted cellular amino acid pools. Inhibition of complex IV also decreased CNTF-dependent elevation of lipid biosynthetic pathway intermediates (Fig. 7c). Quantification of ATP levels revealed that in the wild type retina, 87% of ATP production relied on oxidative phosphorylation (OXPHOS), whereas in the rds mutant only 65% of ATP production was

mitochondrial-dependent (Fig. 7e), suggesting that the degeneration condition itself had led to a decreased dependency on OXPHOS. In CNTF-treated rds mutant, azide caused a 62% reduction of ATP, indicating that a significant portion of the total ATP, about 50% of the wild type level, was not generated through mitochondrial respiration (Fig. 7e). Inhibition of complex IV by azide resulted in 80% reduction of acetyl-CoA in the wild type, but 66% reduction of acetyl-CoA and malate in CNTF-treated rds retinas (Fig. 7f, g). Moreover, analysis using azide showed that mitochondrial respiration was responsible for 50 and 56% of cellular GSH in the wild type and CNTF-treated rds retinas, respectively (Fig. 7m).

The glycolytic pathway enzyme lactate dehydrogenases (LDH) catalyze the conversion between pyruvate and lactate. Inhibition of LDH by GSK2837808A in the wild type retina reduced ATP level by 58%, whereas CNTF-treated rds retinas sustained 65% ATP reduction (Fig. 7h), indicating that aerobic glycolysis contributed to a

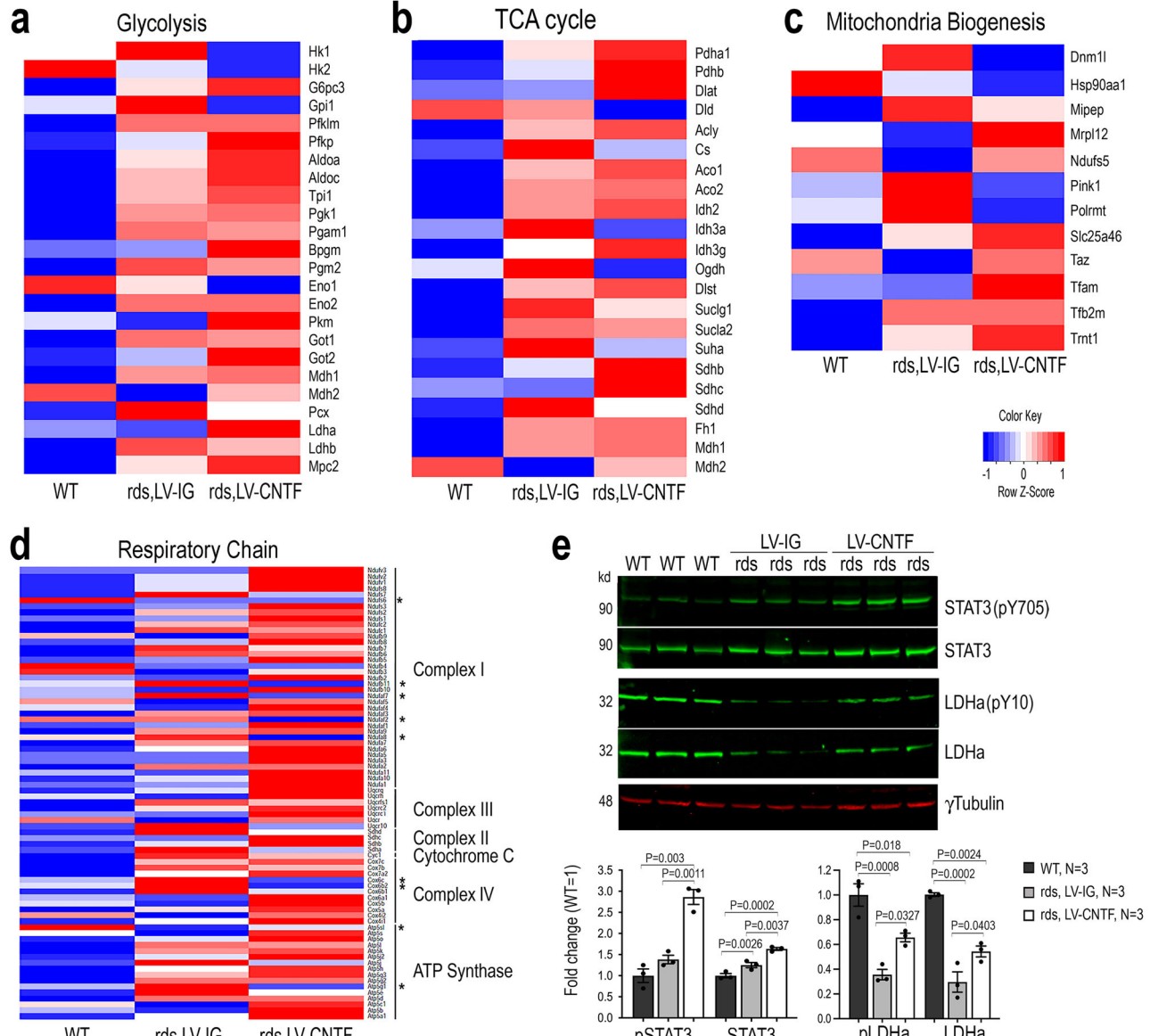

**Fig. 6 | Influence of CNTF on metabolic gene expression and enzyme activity.**
**a**–**d** Transcriptome analysis of WT retina and rds retinas treated with LV-IG or LV-CNTF from P25-P35. Heatmaps show relative transcript levels (average of $N=3$, each $N$ contains two retinas). **a** Glycolytic pathway enzymes and Mpc2. **b** Mitochondrial Pdha1, Pdhb, and TCA cycle enzymes. **c** Participants of mitochondrial biogenesis. **d** Nuclear genome encoded mitochondrial respiratory chain components. Asterisks indicate genes with decreased transcripts compared to WT. **e** Western blot analysis of retinal extracts from WT retina and rds retinas treated with LV-IG or LV-CNTF from P25-P100. Bar graphs show the quantification of CNTF signaling effector pY705 STAT3 and total STAT3, the active pY10 LDHa and LDHa. For **e**, data were presented as mean ± SEM. Independent retinal samples ($N=3$) and adjusted $P$ values based on one-way ANOVA and Tukey all-pairs test are indicated.

larger portion of ATP in CNTF-treated rds retinas. Inhibiting LDH not only increased the AMP to ATP ratios, but also led to increased creatine to p-creatine ratios (Fig. 7d). Furthermore, metabolomics analysis showed significant increases of lactate in CNTF-treated rds retinas compared to rds or rds treated with LV-IG (Fig. 7i), indicating enhanced glycolysis. Consistent with the conversion from lactate into pyruvate to supply mitochondria, inhibiting LDH activity also led to a significant reduction of the TCA cycle product alpha-ketoglutarate (Fig. 7j), likely due to a hindered pyruvate production.

To further examine pyruvate utilization in mitochondrial metabolic processes, we applied the mitochondrial pyruvate carrier inhibitor UK5099, which caused a reduction of acetyl-CoA (Fig. 7k) and TCA cycle intermediate citrate (Fig. 7l), confirming that pyruvate transport to mitochondria supported more than 50% of the citrate production in both the wild type and rds mutant.

## Utilization of fatty acid oxidation by wild type and degenerating retinas

Since the neural retina has highly active lipid biosynthesis and turnover[73], we examined the wild type and rds mutant retinas in their abilities to utilize palmitate as a fuel source in the absence of glucose in the medium (Fig. 8a and Source Data for Fig. 8). CNTF-treated rds retinas showed higher than wild type levels of amino acids using palmitate (Fig. 8b). Inhibiting the carnitine palmitoyltransferase 1 (CPT1) with etomoxir resulted in a marked reduction of amino acids in both the wild type and CNTF-treated rds retinas (Fig. 8b), revealing that the retina relied substantially on fatty acid oxidation under the glucose deprivation condition. As expected, CPT inhibition led to increasing levels of carnitine and reduced levels of acetylcarnitine in both wild type and CNTF-treated rds retinas, (Fig. 8c, i), reflecting disruption of fatty acid transport from the cytoplasm into mitochondria. Compared with the wild type, the rds retina exhibited a nearly two-fold capacity to

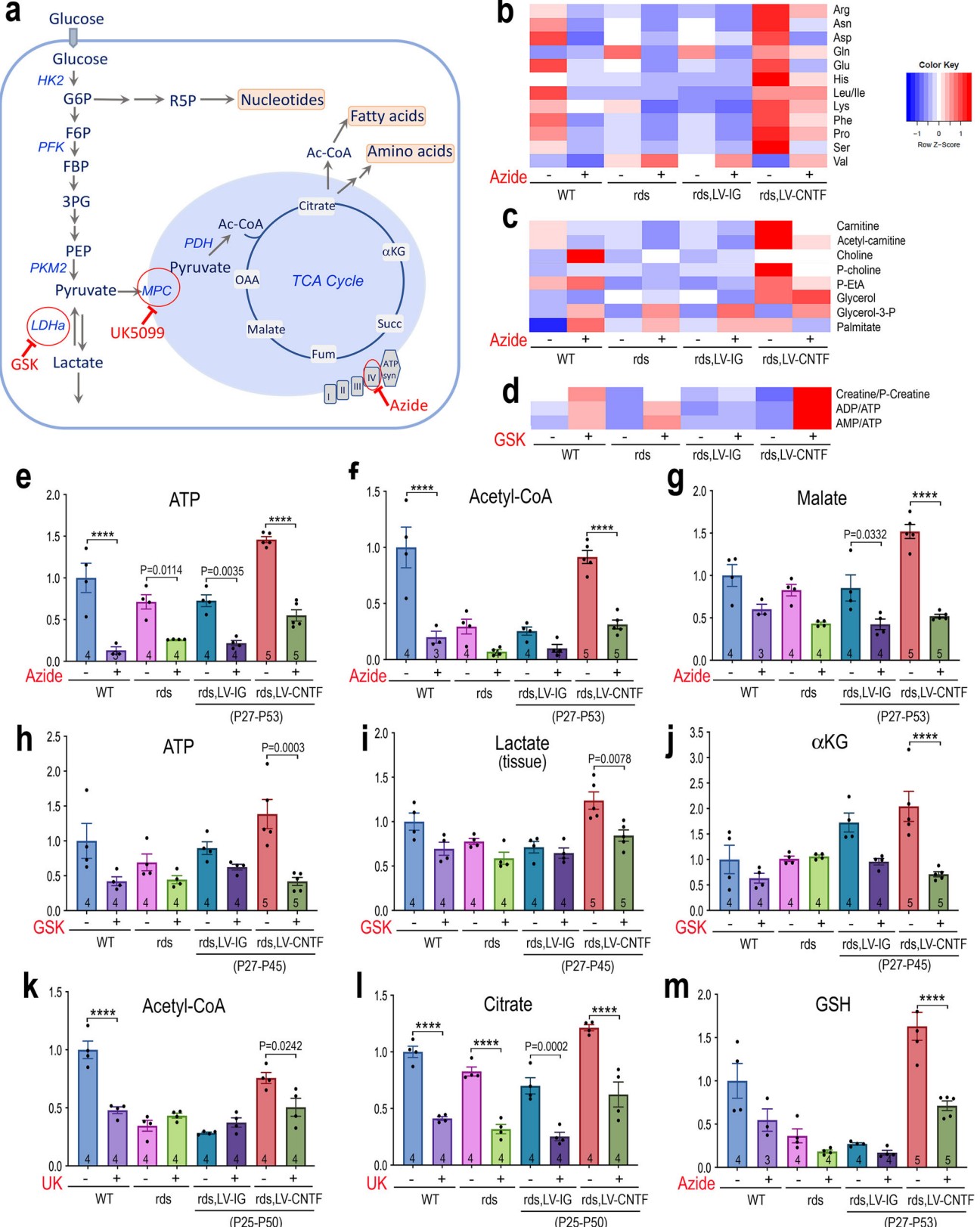

**Fig. 7 | Contribution of glycolysis and mitochondrial activities to retinal metabolism and CNTF-induced metabolic changes. a** Schematic illustration of the glycolytic pathway and TCA cycle with glucose as a fuel. Enzymatic steps affected by specific inhibitors are indicated (red circle). **b–d** Metabolomics heatmaps show the effects of inhibitors in WT, rds, and rds retinas treated with LV-IG or LV-CNTF on metabolites. **b** amino acids. **c** fatty acid synthesis intermediates. **d** energy currency-related metabolites. **e–m** Bar graphs show the effects of inhibitors on the levels of

ATP (**e**, **h**), lactate in the tissue (**i**), acetyl-CoA (**f**, **k**), TCA cycle metabolites (**g**, **j**, **l**), and GSH (**m**). Viral vector treatment periods are shown below treated rds samples. For **e–m**, data were presented as mean ± SEM. Independent sample numbers (*N*) are indicated within the bars. Two-way ANOVA and Tukey's multiple comparison test were applied to the entire group. For clarity, only significant adjusted *P* values for pairs of samples with and without a given inhibitor are shown with *P* < 0.0001 indicated as \*\*\*\*. See Source Data for the entire statistical analysis.

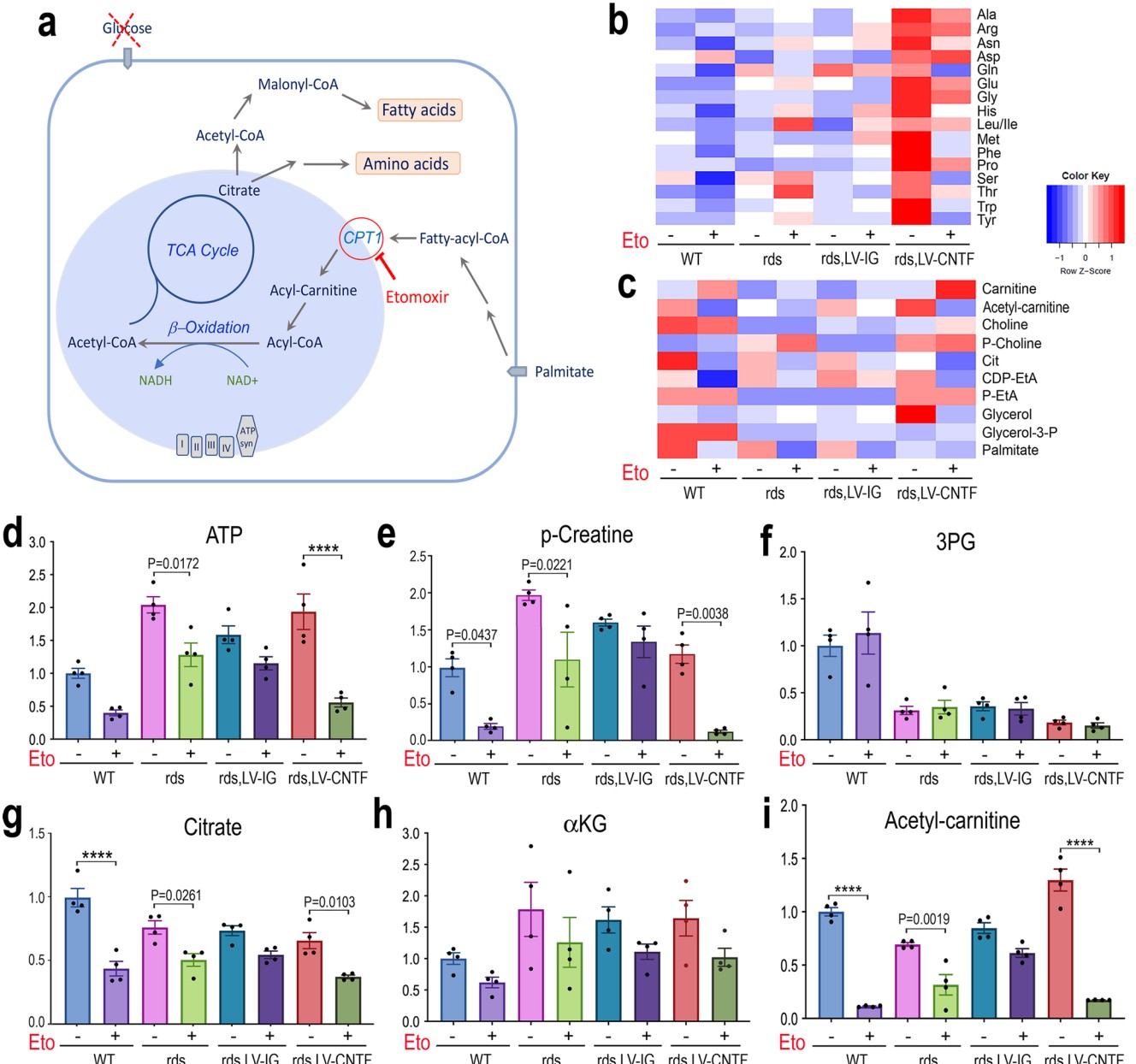

**Fig. 8 | Consumption of fatty acid fuel by wild type and rds mutant retinas.**
**a** Schematic illustration of fatty acid metabolic pathways and Cpt1 as a target of inhibitor Etomoxir. **b**, **c** Metabolomics heatmaps show the effects of Etomoxir on levels of **b** amino acids and **c** fatty acid synthesis intermediates in WT, rds, and rds retinas treated with LV-IG or LV-CNTF from P28-P53. **d**–**i** Bar graphs show the effects of Etomoxir on levels of ATP (**d**), p-creatine (**e**), 3PG (**f**), citrate (**g**), αKG (**h**), and acetylcarnitine (**i**). Independent samples $N = 4$ for all conditions. For **d**–**i**, data were presented as mean ± SEM. Two-way ANOVA and Tukey's multiple comparison test were applied to the entire group. For clarity, only significant adjusted P values for pairs of samples with and without Etomoxir are shown with $P < 0.0001$ indicated as ****. See Source Data for the entire statistical analysis.

produce ATP when palmitate was supplied (Fig. 8d). CNTF-treated rds retina retained the efficiency to synthesize ATP using palmitate, but sustained more severe deficits with etomoxir treatment (Fig. 8d). CPT1 inhibition also nearly demolished p-creatine generation in the wild type and CNTF-treated rds retinas (Fig. 8e), confirming the potential contribution of fatty acid beta-oxidation to the retinal energy buffer. CPT1 inhibition did not affect glycolytic pathway metabolites such as 3PG (Fig. 8f), but did reduce the level of citrate, which can be driven by fatty acid beta-oxidation (Fig. 8g).

## Discussion

In this study, we provide evidence that delivery of the same CNTF used in human trials to a mouse model of retinitis pigmentosa results in a significant change in retinal metabolism, thus revealing a previously unknown cellular mechanism underlying the potent and broad neurotrophic effects of CNTF.

Cellular respiration assays using retinal tissues detected early and persistent decreases in oxygen consumption and elevation of extracellular acidification in the rds mutant. These changes likely reflect the adaptation of the retinal metabolism under the condition of progressive photoreceptor loss. SIM imaging revealed fragmental and ectopically distributed rod mitochondria, suggesting a possible decline of mitochondrial functionality in the rds mutant. Both SIM and TEM analyses showed that CNTF treatment partially restored the morphology of mitochondria in the rod inner segment, the main cellular biosynthetic site for photoreceptors. However, instead of

improving mitochondrial respiration, our results indicated that CNTF treatment caused a reduction of ATP-linked OCR. Furthermore, direct measurements of mitochondrial respiratory chain activities confirmed the suppression of complex I and complex IV functions following CNTF treatment. Concomitant with the suppression of respiratory chain activity, we detected significantly increased ECAR following CNTF treatment, suggesting the likelihood of enhanced glycolysis.

Our biochemical and molecular analyses indeed support the above conclusion and provide a comprehensive view of the retinal metabolic status in the rds mutant and under the influence of CNTF. Metabolomics analyses using glucose as a fuel revealed deficiencies in ATP, amino acids, and fatty acid biosynthetic intermediates, as well as a severe reduction in the major antioxidant glutathione in the rds mutant retina. CNTF treatment effectively improved the retinal energy supply resulting in more than a 50% increase in ATP and a three-fold increase of p-creatine to the wild type levels. In addition, we detected a significant elevation of glycolytic pathway intermediates, TCA cycle metabolites, as well as various anabolic metabolites, including amino acids, lipid biosynthetic intermediates, and nucleotide derivatives. Strikingly, CNTF treatment also restored the important antioxidant glutathione to the level found in wild type retinas. This is not surprising as glutathione biosynthesis requires amino acids glutamate, glycine, and cysteine[74]. Consistent with the metabolomics data, molecular analyses also revealed increased expression of the glycolytic pathway and TCA cycle gene transcripts, and active forms of glycolytic pathway enzymes. Taken together, these results provide strong evidence that CNTF signaling in the degenerating retina leads to a global metabolic alteration by promoting aerobic glycolysis and anabolism. Since metabolomics and transcriptome analyses performed used the entire retina, the results described reflect the summation of responses by various retinal cell types to CNTF rather than photoreceptors alone. Future studies combining genetics and multi-omics approaches are necessary to determine cell type-specific responses to exogenous CNTF.

Metabolomics analyses using glucose in conjunction with metabolic inhibitors permitted us to determine the contributions of glycolysis and mitochondrial respiration to the energy supply of healthy and degenerating retinas. Our results showed a reduced reliance on mitochondrial respiration in CNTF-treated rds retina to generate high levels of ATP and p-creatine, thus further supporting the role of CNTF-dependent elevation of aerobic glycolysis. Consistent with the elevated ECAR in CNTF-treated rds mutant, metabolomics analysis detected elevated lactate in retinal tissues. Further, LDHa transcripts and the active form of LDHa are both increased under CNTF influence. Since LDH catalyzes a bidirectional reaction, it is conceivable that CNTF may have promoted the conversion of lactate toward pyruvate. Notably, the lactate-to-pyruvate enzymatic reaction also produces NADH for cellular metabolism and as an electron donor in oxidative phosphorylation. Indeed, inhibition of the LDH enzymatic activity resulted in a 58% reduction of ATP and a 65% decrease of alpha-ketoglutarate, indicating that LDH-catalyzed pyruvate production was critical for CNTF-dependent energy production as well as the mitochondrial TCA cycle. Consistently, inhibiting the mitochondrial pyruvate carrier led to a significant reduction of acetyl-CoA and citrate. Moreover, in the presence of the complex IV inhibitor azide, we detected not only reduced TCA cycle products, but also amino acids and lipid biosynthetic intermediates, as the production of these anabolic metabolites relies on the supply of acetyl-CoA and the TCA cycle metabolites. These findings demonstrate that CNTF not only upregulates aerobic glycolysis and dampens mitochondrial respiratory chain activity, but also strongly impacts TCA cycle activities that are critical for amino acid and lipid biosynthesis.

It is known that in the case of retinitis pigmentosa, rod photoreceptor death leads to cone cell starvation of glucose[54,55]. In the absence of glucose supply, we found that the rds retina displayed an increased capacity to utilize palmitate to produce ATP and p-creatine compared to the wild type. Blocking palmitate transport into the mitochondria resulted in decreases in ATP and p-creatine production in both wild type and CNTF-treated rds retinas. Interestingly, when provided with palmitate, CNTF-treated rds retinas showed only increased amino acid contents without elevations of other anabolic metabolites compared to using glucose as a fuel. Consistent with retinal consumption of fatty acids through beta-oxidation and the TCA cycle, inhibition of palmitate transport also caused the reduction of citrate, amino acids, and most lipid biosynthetic pathway intermediates, but had no effect on the levels of glycolytic pathway metabolites. Therefore, compared to the wild type, the rds mutant retina can more efficiently use fatty acids as a fuel source when experiencing glucose deprivation.

The CNTF-triggered retinal metabolic changes show striking similarities to the metabolic signatures of cancer cells as described by Warburg nearly a century ago[75]. We have demonstrated previously that CNTF primarily activates the Jak-STAT and ERK pathways in the retina[38–40]. Activation of STAT3 has long been deployed to maintain the pluripotency of stem cells, which rely heavily on aerobic glycolysis to supply energy and various metabolites for proliferation, epigenetic modification, and cellular anabolic processes[76,77]. Elevated STAT3 signaling has been associated with the oncogenic process[78–80]. Intriguingly, in addition to the nuclear-localized pY705 STAT3 dimer to regulate target gene transcription, the Ser727 phosphorylated STAT3 has been shown to enter mitochondria and enhance electron transport chain function, as well as promote oncogenic transformation[81,82]. However, the exact role of pS727 STAT3 in mitochondrial function remains to be resolved[83–85]. A previous study has shown that elevating wild type or a constitutively active STAT3 in mutant mouse rod photoreceptors can delay degeneration and that the pY705 STAT3 mediates this protective effect[86]. Since CNTF signaling asserts a strong influence on the retinal transcriptome[44], the precise functions of pY705 and pS727 STAT3 in CNTF-mediated metabolic modulation require further investigations. Consistent with our results that exogenous CNTF suppresses mitochondrial respiratory chain activity and promotes cell viability, accumulating evidence indicates that partial uncoupling of mitochondrial respiration can reduce oxidative stress and support neuronal survival under stressful conditions[87–91]. Furthermore, blocking mitochondrial transport of pyruvate has been shown to increase glycolysis and potentiate endogenous stem cell activities[92], as well as attenuate neuronal loss[93]. It may appear paradoxical that CNTF improves mitochondrial morphology on the one hand, yet suppresses respiratory chain function. However, it is plausible that CNTF signaling results in a partial uncoupling, but does not compromise the TCA cycle activity in the mitochondrial matrix, thus enhancing anabolism.

The results of this study provide much-needed insight at the molecular and biochemical levels for the ongoing clinical trials aimed at treating blinding diseases. Recent studies have shown that MacTel type 2, which causes late-onset retinal degeneration, is associated with metabolic dysfunction in the serine-glycine biosynthetic pathway[94,95]. In addition, metabolic changes in phospholipids, including phosphatidylethanolamines, have been detected among MacTel patients[96]. Based on our findings, one possible explanation for the observed efficacy of the phase II CNTF trial for MacTel type 2[31] is likely due to CNTF-dependent alterations of retinal metabolism, especially the elevation of retinal amino acid contents. CNTF is known to have neurotrophic effects on RGC survival and RGC axon regeneration[13,19]. Future investigations examining retinal cell type-specific effects of CNTF will facilitate elucidating cellular mechanisms for the ongoing CNTF glaucoma trials. In summary, the findings of this study have revealed

cellular responses of the neural retina to CNTF treatments, thus suggesting potential therapeutic treatments for neurodegenerative diseases.

## Methods

### Animals

Mice were kept on a 12/12-h light/dark cycle with Rodent Diet 20 (Pico-Lab, catalog number 15053). The rds transgenic mice carrying the Prph2(P216L) mutation[64] were maintained by crossing with the wild type CD1 mice from the Jackson Laboratory. Progeny of the cross without the Prph2(P216L) transgene were used as wild type littermate controls. Mice expressing rod photoreceptor-specific mitochondrial PhAM reporter were generated by crossing mice that carry both Rho iCre75[65] (Jackson Laboratory Stock No.015850) and the rds/Prph2(P216L) transgenes with the PhAM reporter mice[66] (Jackson Laboratory Stock No.018385). Genotyping was performed by tail genomic DNA extraction followed by PCRs using primers listed in the Supplementary Table. Equal numbers of male and female mice are used for a given assay. All animal procedures followed National Institutes of Health guidelines and were approved by the Animal Research Committee of University of California Los Angeles under protocol number ARC-1996-047.

### Lentiviral vector production and intraocular delivery

The lentivirus LV-CNTF and the control lentivirus LV-IG share the same vector backbone and encode the CMV promoter followed by CNTF-IRES-GFP or IRES-GFP, respectively[40]. The LV-CNTF expresses the same secreted form of human CNTF with S166D and G167H substitutions used in clinical trials[42]. The lentiviral vectors and helper plasmids were used to co-transfect HEK293t cells, and the media were collected 48 h post-transfection. Ultracentrifugation-concentrated viral particles were resuspended and used to obtain viral titers by serial dilution and infection of HEK293t cells followed by immunolabeling and quantification of GFP-positive clones[40,97]. For in vivo delivery, lentiviral stocks with titers of $1 \times 10^7$ CFU/ml were injected subretinally at 0.5 µl per eye.

### Confocal, super-resolution, and transmission electron microscopies

Cells or retinal tissues were fixed with 4% (wt/vol) paraformaldehyde in PBS and processed as described previously[39]. Confocal fluorescent images were captured using Olympus FluoView 1000 scanning laser confocal microscope. Super-resolution images were captured with either Zeiss Airyscan LSM 800 or General Electric DeltaVision OMX microscope for structure illumination microscopy (SIM) using PlanApoN 60x/1.42 NA oil objective (Olympus). For Airyscan imaging, the retina was dissociated using papain as described[98]. For SIM imaging, 14-µm thickness cryosections were adhered to #1.5 coverslip (Thermo Fisher Scientific) coated with Matrigel (BD Biosciences and StemCell Technologies) and mounted on a glass slide using Vectashield mounting medium (Vectorlabs). The coverslip/slide was sealed with CoverGrip (Biotium). Images were acquired in 3D-SIM mode using a Z-spacing of 0.125 µm and reconstructed using Softworx software (GE Healthcare). Imaris software (Oxford Instruments) was used to extract 5-µm thickness 3D images and create 3D rotating video clips. For TEM, eyes were fixed in 2% (wt/vol) formaldehyde and 2.5% (wt/vol) glutaraldehyde in 0.1 M sodium phosphate buffer and processed as described in ref. 99. TEM images were acquired using JEM-1400 (JEOL) electron microscope.

### Mitochondrial morphology analysis

SIM images were analyzed in FIJI/ImageJ[100] using the open-source software plugin MitoMap (http://www.gurdon.cam.ac.uk/stafflinks/downloadspublic/imaging-plugins)[68]. The plugin automates the process to compute mitochondrial volume (µm³), surface area (µm²), and other geometrical features in a region of interest (ROI). For each sample analyzed, the 32-bit OMX images stack (dv file) are loaded onto ImageJ using the Bio-formats plugin[101], and an ROI was chosen that includes the photoreceptor inner segment ($40 \times 15 \times 10$ µm³) region. The MitoMap plugins converted the images stack to 16-bit and applied Otsu thresholding[102] to extract the PhAM reporter-labeled mitochondrial volume. Before the quantification analysis, objects with volumes smaller than 0.1 µm³ were excluded to eliminate artifacts. The principal component analyses were performed on R Studio using "factoextra" and "FactoMineR" packages[103].

### Seahorse assays

For tissue OCR and ECAR assays, retinas were dissected in HBSS and tissue disks containing all retinal layers were obtained using a 1 mm diameter biopsy puncher (Miltex)[104]. Retinal disks were washed with unbuffered Seahorse XF Base DMEM pH 7.4 (Agilent) and placed in a well of 96-well microplate containing 175 µl medium supplemented with 6 mM glucose (pH 7.4). Retinal disks were incubated for 30 min at 37 °C prior to running the assay using a Seahorse XF96 Analyzer (Agilent). During the assay, 3 µM oligomycin (Port A), 0.6 µM and 1.1 µM FCCP (Ports B and C), and 2 µM antimycin A and 2 µM rotenone (Port D) were injected into the assay medium (Supplementary Table).

For isolated mitochondria seahorse assays, eyes were dissected in cold PBS and 4 retinas were pooled for each sample (N). After washing and removal of PBS, 500 µl of MSHE buffer (210 mM mannitol, 70 mM sucrose, 5 mM HEPES, 1 mM EDTA, 0.5% BSA, pH 7.2) was added to the pooled retinas. Retinal tissues were homogenized by drawing through a 23 G needle 20 times, followed by centrifugation at 800×$g$ for 10 min at 4 °C. The supernatants containing mitochondria were collected and transferred to a new tube, and the pellets were homogenized and centrifuged again to re-collect the supernatants. The supernatants were then combined and centrifuged at 8000×$g$ for 10 min at 4 °C. The resulting pellets containing mitochondria were resuspended in 800 µl of MSHE and centrifuged again at 8000×$g$ for 10 min at 4 °C. After discarding the supernatant, the mitochondria pellet was resuspended in 35 µl of mitochondrial assay solution (MAS; 220 mM mannitol, 70 mM sucrose, 10 mM potassium phosphate monobasic, 5 mM magnesium chloride, 2 mM HEPES, 1 mM EGTA, pH 7.2) without BSA. Mitochondrial protein contents were measured using the BCA assay (Pierce). Isolated mitochondria were loaded into an ice-cold Seahorse XF96 microplate with 15 µl of 10x substrate solution (50 mM pyruvate and 50 mM malate, or 50 mM succinate and 20 µM rotenone). For Complex I-driven respiration (pyruvate + malate), 7 µg protein was loaded per well; while 4 µg protein/well was loaded for Complex II-driven respiration (succinate + rotenone). The volume was adjusted to 20 µl per well and the mitochondria plate was centrifuged at 2100×$g$ for 5 min to allow mitochondria to adhere to the bottom of the well. After centrifugation, the total volume of the well was adjusted to 130 µl with ice-cold MAS with 0.1% fatty acid-free BSA (pH 7.2 adjusted with 1 M KOH). During the assay, compounds were injected from the ports of the XF96 Analyzer (Supplementary Table). The isolated mitochondria conditions include injection of: (1) 4 mM ADP in the presence of pyruvate and malate or succinate and rotenone (State 3: maximal ATP synthesis capacity), (2) 3 µM of the ATP Synthase inhibitor, oligomycin (State 4o: proton leak), (3) the chemical uncoupler, FCCP at 4 µM, and (4) 2 µM of the Complex III inhibitor, antimycin. When measuring Complex IV respiration, isolated mitochondria conditions include injection of (1) 4 mM ADP in the presence of pyruvate and malate or succinate and rotenone (State 3: maximal ATP synthesis capacity), (2, 3) 1 mM TMPD to (donate electrons to cytochrome c/Complex IV) with ascorbate to keep TMPD in the reduced state, and (4) 40 mM of the Complex IV inhibitor, azide.

## Metabolomics analysis

For polar metabolite extraction from retinal tissues, retinas were dissected in cold Krebs-Ringer Bicarbonate Buffer pH 7.4 (KRB: 119.78 mM NaCl, 2.6 mM CaCl$_2$, 4.56 mM KCl, 0.49 mM MgCl$_2$, 0.7 mM Na2HPO$_4$, 1.3 mM NaH$_2$PO$_4$, 14.99 mM NaHCO$_3$, and 10 mM HEPES) with 5 mM substrates (glucose or palmitate). Whole retina or 1 mm retinal disks were incubated in 500 µl of KRB with or without inhibitors (Supplementary Table), for 60 min at 37 °C. After removal of the KRB medium, retinal tissues were homogenized in 500 µl of cold 80% methanol by drawing through a 23G needle 20 times. The tissue extracts were spun in a microfuge at 13,000 rpm at 4 °C for 10 min. The supernatants were transferred into glass vials, 2 nmoles of norvaline was added to each vial, and the extracts were desiccated using an EZ-2Elite evaporator and stored at −80 °C.

Dried metabolites were resuspended in 50% ACN:water and 1/10th was loaded onto a Luna 3 µm NH2 100 A (150 × 2.0 mm) column (Phenomenex). The chromatographic separation under HILIC conditions was performed on a Vanquish Flex (Thermo Scientific) with mobile phases A (5 mM NH4AcO pH 9.9) and B (ACN) and a flow rate of 200 µl/min. A linear gradient from 15% A to 95% A over 18 min was followed by 9 min isocratic flow at 95% A and re-equilibration to 15% A. Metabolites were detected with a Thermo Scientific Q Exactive mass spectrometer run with polarity switching (+3.5/−3.5 kV) in full scan mode with an m/z range of 65–975. TraceFinder 4.1 (Thermo Scientific) was used to quantify the targeted metabolites by area under the curve using expected retention time and accurate mass measurements (<5 ppm). Data analysis was performed using in-house R scripts (https://github.com/graeberlab-ucla/MetabR).

## RNA-sequencing

The whole retinal transcriptome analysis was performed as described[44]. Briefly, total RNA was isolated from the wild type, and rds/Prph2(P216L) mutant retinas treated with LV-IG or LV-CNTF from P25-P35 (N = 3, each N contains two retinas). Sequencing libraries were prepared using the Illumina TruSeq Stranded Total RNA with Ribo-Zero Gold Library Prep kit. Paired-end sequencing of 69 bp of the libraries was performed using HiSeq 4000 Sequencer (Illumina). After quality control and filtering, the median size was 5.65 Gb per library (range 3.66–7.26 Gb). Sequenced reads were then aligned to the Mouse reference mm10 from UCSC (genome-euro.ucsc.edu) using Hisat2 (v2.0.4)[105]. The expression levels were normalized by calculating the fragments per kilobase million reads (FPKM) values.

## Western blots

For western blot analysis, the dissected retina was washed with PBS, then lysed with RIPA (50 mM Tris, 150 mM NaCl, 1% Triton X-100, 2% BSA, and complete protease inhibitor cocktail, pH 7.4). The lysates were then resolved on SDS-PAGE and western blots were performed as described in ref. 40. The protein signals were imaged using Odyssey® CLx Imaging System (LI-COR Biosciences). Primary and secondary antibodies used are summarized in the Supplementary Table.

## Statistics

All N numbers represent independent samples analyzed, as indicated pertinently in each figure, including Supplementary Figures. All error bars in bar graphs of figures and supplementary figures are presented as mean value ± SEM. For Fig. 3a, a two-tailed student's t-test was used to compare the wild type with rds mutant data (Source Data for Fig. 3). For the rest of the data in various figures, one-way or two-way ANOVA and Tukey's multiple comparison tests, when appropriate, were performed (see Source Data for detailed statistics for each figure). Actual P values are shown in figures with $P < 0.05$ considered statistically significant and $P < 0.0001$ indicated as ****.

## Reporting summary

Further information on research design is available in the Nature Research Reporting Summary linked to this article.

## Data availability

RNA-sequencing data have been deposited at Gene Expression Omnibus (GEO) and will be publicly available with Accession number GSE216208. All data generated for this study are included in the main and supplementary figures. For all quantitative figures, data of individual values, as well as the results of statistical tests, are provided in the Source Data files with the paper. Other data that support the findings of this study are available on request from the corresponding author. Source data are provided with this paper.

## Code availability

Images analysis of mitochondria used FIJI/ImageJ[97] open-source software plugin MitoMap (http://www.gurdon.cam.ac.uk/stafflinks/downloadspublic/imaging-plugins)[68]. Code used for metabolomics data analysis is deposited in Github Repository: https://github.com/graeberlab-ucla/MetabR.

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

## Acknowledgements

This work was supported by NIH grant R01EY026319 to X.-J.Y., NIH core grant P30EY000331, and an unrestricted grant from the Research to Prevent Blindness to the Department of Ophthalmology at University of California Los Angeles.

## Author contributions

K.D.R and X.-J.Y., conception and research designs; K.D.R and T.T.T.N., SIM and MitoMap analysis; D.B., electron microscopy; K.D.R. and Y.W., transcriptome analysis; L.S. and L.V., mitochondrial respiration assays; K.R. and O.S., data interpretation; K.D.R. and X.Z., Western blot analysis; J.t.H., metabolomics; K.D.R. and X.-J.Y., data analysis and figure preparation; X.-J.Y., manuscript writing.

## Competing interests

The authors declare no competing interests.
