## [Peer Review File · Nature Communications]

Ciliary neurotrophic factor-mediated neuroprotection involves enhanced glycolysis and anabolism in degenerating mouse retinasREVIEWER COMMENTS

Reviewer #1 (Remarks to the Author):

This is an interesting and exciting study that details mechanisms of neuroprotection of CNTF on retina metabolism in a mouse model of retinitis pigmentosa. The authors show improved morphology of photoreceptor mitochondria in the Prph2 model of retinal degeneration. CNTF treatment resulted in elevated extracellular acidification rate, as a surrogate for glycolysis. Further, CNTF treatment increased energy production, aerobic glycolysis, and cellular contents for most amino acids. RNA-sequencing showed an increase in glycolysis pathway transcripts. The data is of interest as it provides a mechanistic insight for understanding the role of CNTF in neuroprotection in retinal degenerative diseases. I would be supportive of publication in Nature Commun if the authors address the following points.

1. The human clinical trials for retinitis pigmentosa are not well described. An important point would be for the authors to include further details of CNTF human clinical trials. (Birch DG, Weleber RG, Duncan JL, Jaffe GJ, Tao W; Ciliary Neurotrophic Factor Retinitis Pigmentosa Study Groups. Randomized trial of ciliary neurotrophic factor delivered by encapsulated cell intraocular implants for retinitis pigmentosa. *Am J Ophthalmol.* 2013 Aug;156(2):283-292) showed that long delivery of CNTF through an encapsulated cell implant did not show a therapeutic benefit in patients with retinitis pigmentosa. It would be important for the authors to clarify that the benefit seen in mouse does not necessarily translate to patients suffering from retinitis pigmentosa.

2. CNTF alters the rod photoreceptor mitochondria; however, metabolomics analysis and transcriptomic analysis are all performed on the whole retina. As CNTF increases survival of other neurons such as retinal ganglion cells, the authors need to discuss whether the cell-type effects seen from CNTF on aerobic glycolysis and anabolic activities are through rod photoreceptors or could be through other cell-types in the retina such as retinal ganglion cells?

Reviewer #2 (Remarks to the Author):

Herein the authors investigate the metabolic rewiring that occurs in a mouse model of RP with and without lentiviral expression of CNTF, a therapeutic currently in clinical trials but whose mechanism of action is not fully established.

The authors present a nice report with multiple analytical approaches to probing and profiling metabolism, however, some improvements are necessary to reach the high level of Nature journals. My questions and concerns are listed below:

1) The P timing is not clear. My background is biochemistry/metabolism and thus I find the introductory material about RP and photoreceptors to be highly valuable and mostly new to me. The results section begins to use nomenclature like P17, P35 without explanation of its meaning and why such timing or aging is important.

2) An expansion of the importance of phosphocreatine is important and needed. It is not widely thought of as an obvious energy currency (akin to ATP), and certainly the ophthalmology readership would benefit from slightly more information here.

3) There seems to be a conflict between the observed decreases in OCR juxtaposed with increased TCA cycle metabolites citrate, aKG, malate. Is there any information on succinate and fumarate? The authors should comment on this evidence, which seems to be conflicting, or add experimentation to probe these differential observations (i.e. is the TCA cycle becoming uncoupled from the ETC?).

4) a supplementary table should be included that reports all observed metabolites with peak area

measurements for each sample. It is hard to argue anabolism without ribose phosphate, nucleotides and nucleosides, etc.

5) In general, the metabolomics data presentation seems to focus only on a relative handful of metabolites that are changing rather than presenting a full context of pathways. It is presumptive to draw conclusions about glycolysis from the measurements of only phosphoglycerate and phosphoenolpyruvate, same for the TCA cycle, and other related pathways (glutaminolysis, anaplerosis) are unmentioned. I would recommend an unsupervised analysis of the metabolomics data first - like PCA, PLS-DA (partially supervised), etc to see the most prominent changes or pathways then follow up with the types of plots shown currently.

6) Figure 5 - ATP and GTP are shown, but what about other nucleotides? Evidence for methylthioadenosine seems weak/without context. Plots in this figure are not aligned. Heat maps would benefit from hierarchical clustering so that metabolites that trend similarly are positioned closely... as of now it appears to be alphabetical order or random order which do not add scientific meaning. Here and in general, many pertinent metabolites are missing - lactate, pyruvate, glucose, hexose phosphate, ribose phosphate, glyceraldehyde phosphate, succinate, fumarate.

7) Figure 5 - GSH and GSSG have the same trend. Have the authors explored de novo GSH synthesis (either enzymes or metabolites) or NADPH levels (glutathione reductase)?

8) Figure 6 - hierarchical clustering needed in heat maps

9) Figure 8 - "Ac-carnitine" is this meant to be "acylcarnitine" as in the legend, which is a very general name, or "acetylcarnitine" as in the C2 carnitine? Either way, more information is needed, like levels of palmitoylcarnitine. C2 carnitine does not need CPT to cross the mitochondrial membrane and is not a sufficient readout of carnitines at large.

10) typo in line 14

11) For a general readership like Nature Communications, the title needs to be flashier and CNTF should be defined.

Reviewer #3 (Remarks to the Author):

The manuscript NCOMMS-21-42240, "CNTF-mediated neuroprotection involves enhanced glycolysis and anabolic activities in degenerating retinas" is a comprehensive metabolomics study of the effects of ciliary neurotrophic factor (CNTF) in the rds mouse model of retinal degeneration. This work represents an extension of the authors' previous work on CNTF-mediated protection of photoreceptors (Rhee et al., 2013) and provides much-needed mechanistic insight into the biology underlying CNTF-mediated neuroprotection that is relevant to current human clinical trials.

The authors are to be commended for their detailed investigation of the metabolic pathways underlying CNTF-mediated protection in the rds mouse. They provide convincing evidence that CNTF has extensive effects on the metabolic state of the degenerating retina including suppression of mitochondrial respiratory chain function, promotion of aerobic glycolysis, and improvement in the retinal energy supply. Furthermore, they utilize robust metabolomics analyses to determine the contribution of glycolysis and mitochondrial respiration in healthy vs. degenerating retinas and suggest that CNTF-mediated protection is correlated with increases in aerobic glycolysis.

Comments:

1. CNTF was delivered via subretinal injection; it would be helpful to comment on the location and level of CNTF expression in retinal tissue to correlate with the histologic findings. I.e. are the images shown in Fig 1A-C at the site of the subretinal bleb, or were mitochondrial morphological changes

seen diffusely in all retinal sections due to the secreted nature of CNTF? Were there any local inflammatory or histologic changes seen?

2. How were the timepoints for LV-IG or LV-CNTF injection, harvesting, and various analyses determined, and do the metabolic effects of CNTF change if delivered in early vs. late-stage disease? Given previous reports of decreased visual function after prolonged exposure to high levels of CNTF, were any additional time points assessed to determine if the metabolic profiles reported change over time? Were similar experiments conducted in more rapid murine models of retinal degeneration than the rds mouse and if so, did CNTF have a similar effect on the metabolic profile in these models?

3. In Fig 2A, it appears that LV-IG, but not LV-CNTF improves the distribution isotropy. Conversely, in Fig 3B the basal OCR appears to be decreased in retinas treated with LV-CNTF and even lower in retinas treated with LV-IG. These counterintuitive findings should be clarified in the text.

4. Clarity of Fig 6 could be improved by including transcriptome analyses of untreated rds mice, if available, in addition to WT, LV-IG, and LV-CNTF treated groups.

5. The authors' claims would be strengthened with evidence of visual function improvement in LV-CNTF treated rds animals at the time points assessed.

Details of the metabolomics methods utilized in this manuscript are beyond my scope of expertise.

Overall this is a logical, well-done study that provides novel mechanistic insights of CNTF-mediated neuroprotection that are of great relevance to current ongoing human clinical trials.

Reviewer #4 (Remarks to the Author):

Key Results: Ciliary neurotrophic factor (CNTF) is one of the most studied neurotrophic factors in the retina and exhibits potent neuroprotective activity in many models of retinal degeneration, however, mechanisms underlying its actions are incompletely understood. The authors of this paper investigated the influence of CNTF on retinal metabolism in a mouse model of retinitis pigmentosa. Employing a wide array of cutting-edge research technologies, they present a comprehensive evaluation of retinal metabolic status of the rds mutant and the action of CNTF. They show that CNTF improved photoreceptor mitochondrial morphology in the rds PRs. Metabolic flux respiration assays documented an attenuation of mitochondrial respiratory chain activity and increased glycolysis. Metabolomics analyses using glucose as a fuel revealed deficiencies in ATP, amino acids, and fatty acid biosynthetic intermediates, as well as a severe reduction in the major antioxidant glutathione in the rds retina. CNTF signaling exerted a strong influence on retinal metabolism in the rds retina, promoting glycolysis and stimulating anabolism. Transcriptome and metabolomics analyses revealed increased TCA cycle products, lipid biosynthetic pathway intermediates, nucleotides, and amino acids. Importantly, CNTF restored the key antioxidant glutathione to control values in the degenerating retina. This detailed and comprehensive study demonstrates that CNTF profoundly impacts the metabolic status of degenerating retinas by promoting aerobic glycolysis and anabolic metabolism revealing the fundamental cellular mechanism by which a neurotrophic factor promotes neuronal viability in disease conditions. These studies also provide valuable mechanistic insight at the molecular and biochemical levels for the ongoing CNTF clinical trials for blinding diseases.

Validity The data are presented clearly with detailed figures and illustrations. The data interpretation is consistent with the data presented.

Significance: The data presented in this manuscript make significant contributions to the literature from several perspectives. (1) Evidence that CNTF induces a profound change in retinal metabolism in retinal degeneration revealing a key cellular mechanism of cytoprotection. (2) The CNTF-induced shift from mitochondrial respiration to aerobic glycolysis is similar to the Warburg effect observed in cancer cells and pluripotent stem cells providing additional insight into the anabolic and cytoprotective nature of this metabolic shift. (3) Provided much needed mechanistic insights for ongoing clinical trials

of CNTF for blinding diseases.

Data and methodology. An impressive aspect of this manuscript is the use of diverse and complementary methodology to address the questions. The data fit together nicely and the data obtained from the different methodological approaches support each other. Moreover, included in supplemental materials, the authors have included 5 videos of retinal histology, two metabolomic heat map figures, 6 tables containing all of the experimental data and a table of reagents.

Analytical approach. The analytical approach and statistics are valid and consistent with similar studies.

Suggested improvements: This manuscript is suitable for publication as it stands. However, given the effort expended in the mitochondrial morphological analysis (movies) and MitoLoc characterization, additional discussion of the functional implications of the differences in 3D morphology measured in the wild-type, rds and rds+CNTF would be valuable.

Clarity and context The manuscript is clear, organized and well-written. The authors present their research in the context of previous studies conducted by them and by other investigators.

References: The manuscript references previous literature appropriately and comprehensively

RESPONSE TO REVIEWS

We thank the reviewers for recognizing that our study advances the fundamental mechanistic understanding of CNTF-mediated neuroprotection. We hereby address each point raised in their insightful reviews.

*Note: Reviewer's original comments in their entirety are included below in ***Italic*** font.

Reviewer #1

This is an interesting and exciting study that details mechanisms of neuroprotection of CNTF on retina metabolism in a mouse model of retinitis pigmentosa. The authors show improved morphology of photoreceptor mitochondria in the Prph2 model of retinal degeneration. CNTF treatment resulted in elevated extracellular acidification rate, as a surrogate for glycolysis. Further, CNTF treatment increased energy production, aerobic glycolysis, and cellular contents for most amino acids. RNA-sequencing showed an increase in glycolysis pathway transcripts. The data is of interest as it provides a mechanistic insight for understanding the role of CNTF in neuroprotection in retinal degenerative diseases. I would be supportive of publication in Nature Commun if the authors address the following points.

1. The human clinical trials for retinitis pigmentosa are not well described. An important point would be for the authors to include further details of CNTF human clinical trials. (Birch DG, Weleber RG, Duncan JL, Jaffe GJ, Tao W; Ciliary Neurotrophic Factor Retinitis Pigmentosa Study Groups. Randomized trial of ciliary neurotrophic factor delivered by encapsulated cell intraocular implants for retinitis pigmentosa. Am J Ophthalmol. 2013 Aug;156(2):283-292) showed that long delivery of CNTF through an encapsulated cell implant did not show a therapeutic benefit in patients with retinitis pigmentosa. It would be important for the authors to clarify that the benefit seen in mouse does not necessarily translate to patients suffering from retinitis pigmentosa.

We have now added to the introduction a brief description of the RP trial outcomes as reported by Birch et al in 2013. The paper described two CNTF implant trials, CNTF3 for late RP (n=65) and CNTF4 for early RP (n=64). The longest treatment was for 2 years, and neither trial showed therapeutic benefit by accessing the primary outcome, the best-corrected visual acuity (BCVA). The high dose group of CNTF4 showed a reversible decrease of sensitivity. Both studies detected an increase of retinal thickness at the higher dose. These trial results are consistent with studies using various rodent models, including our own, which have shown that constitutive exposure to high levels of CNTF can dampen visual function despite the robust rescue of photoreceptor death. This dichotomy of CNTF effects is part of the motivation for our current study, which aims at elucidating the cellular mechanism underlying the potent neuroprotection by CNTF and its potential detrimental effect on visual function. Findings of our study described here, therefore, provide much needed insight for the ongoing clinical trials for retinal degenerative diseases MacTel type2 and glaucoma using the same encapsulated cell implants as the previous CNTF trials.

2. CNTF alters the rod photoreceptor mitochondria; however, metabolomics analysis and transcriptomic analysis are all performed on the whole retina. As CNTF increases survival of other neurons such as

retinal ganglion cells, the authors need to discuss whether the cell-type effects seen from CNTF on aerobic glycolysis and anabolic activities are through rod photoreceptors or could be through other cell-types in the retina such as retinal ganglion cells?

This is an important point. We have presented data of CNTF's influence on rod photoreceptor mitochondrial morphology in this study. However, the lentiviral expression of the secreted CNTF indeed affects the entire retina, as shown in our previous publication (Rhee et al, PNAS, 2013; Reference 40). The transcriptome and metabolomics analyses we present here were using the entire retina treated with CNTF, thus reflecting the overall impact of CNTF, which are not limited to photoreceptors. We now have emphasized this point in the discussion. In our ongoing investigations, we are focusing on retinal cell type-specific responses to CNTF using single cell RNA-sequencing and metabolic assays of sorted cell types in conjunction with genetic manipulations. These approaches will provide further information on how different retinal cell types respond to CNTF, however, are beyond the scope of this report.

Reviewer #2

Herein the authors investigate the metabolic rewiring that occurs in a mouse model of RP with and without lentiviral expression of CNTF, a therapeutic currently in clinical trials but whose mechanism of action is not fully established. The authors present a nice report with multiple analytical approaches to probing and profiling metabolism, however, some improvements are necessary to reach the high level of Nature journals. My questions and concerns are listed below:

1) The P timing is not clear. My background is biochemistry/metabolism and thus I find the introductory material about RP and photoreceptors to be highly valuable and mostly new to me. The results section begins to use nomenclature like P17, P35 without explanation of its meaning and why such timing or aging is important.

We apologize for not stating the relevant time points used in this study. We have now added a brief statement of the photoreceptor degeneration time course of the *rd5 Prph2(P216L)* transgenic mouse model as well as spelled out that the "P" represents "postnatal day" at the beginning portion of the result section on page 5.

2) An expansion of the importance of phosphocreatine is important and needed. It is not widely thought of as an obvious energy currency (akin to ATP), and certainly the ophthalmology readership would benefit from slightly more information here.

Thank you for mentioning this point. We have now included a statement and references (References 69, 70) regarding the role of phosphocreatine as an energy buffer in neurons.

3) There seems to be a conflict between the observed decreases in OCR juxtaposed with increased TCA cycle metabolites citrate, aKG, malate. Is there any information on succinate and fumarate? The authors should comment on this evidence, which seems to be conflicting, or add experimentation to probe these differential observations (i.e. is the TCA cycle becoming uncoupled from the ETC?).

To demonstrate the influence of CNTF on TCA cycle, we now include in Source Data Figure S2 bar graphs that illustrate TCA cycle related metabolites. CNTF treatment increases citrate, aconitate, α-ketoglutarate, and malate in the metabolomics analysis. Our targeted metabolomics assay does not detect fumarate.

The reviewer raises a very interesting point: our data show increases of TCA cycle metabolites, but also inhibition on the ETC activities. Indeed, CNTF treatment alters the mitochondrial morphology of the rds mutant rod photoreceptors from highly fragmental to more elongated and enlarged; however, the ETC function is partially suppressed. Does this represent an uncoupling of TCA cycle from ETC? Since TCA cycle occurs in the mitochondrial matrix, it is plausible that under ETC suppression the TCA cycle can remain robust in the restored matrix compartment. The partial uncoupling of mitochondrial membrane potential has been reported to promote neuronal survival (References 86-90). It is conceivable that the TCA cycle generated metabolites such as citrate, acetyl CoA, αKG can fuel cellular anabolism. This will be a key issue to be further investigated.

4) a supplementary table should be included that reports all observed metabolites with peak area measurements for each sample. It is hard to argue anabolism without ribose phosphate, nucleotides and nucleosides, etc.

We now provide the complete data set of metabolomics with all observed metabolites using glucose as a fuel as Source Data Supplementary Table 4. We also included bar graphs (Supplementary Fig.2, Supplementary Fig.3, Supplementary Fig.4) for easy visualization of different categories of metabolites relevant to the glycolytic pathway, TCA cycle, energy and redox currency, fatty acid biosynthetic intermediates, amino acids, and nucleotide derivatives.

5) In general, the metabolomics data presentation seems to focus only on a relative handful of metabolites that are changing rather than presenting a full context of pathways. It is presumptive to draw conclusions about glycolysis from the measurements of only phosphoglycerate and phosphoenolpyruvate, same for the TCA cycle, and other related pathways (glutaminolysis, anaplerosis) are unmentioned. I would recommend an unsupervised analysis of the metabolomics data first - like PCA, PLS-DA (partially supervised), etc to see the most prominent changes or pathways then follow up with the types of plots shown currently.

Please see the complete metabolomics data set (Source Data Supplementary Table 4) and featured metabolic pathway data (Supplementary Fig.2, Supplementary Fig.3, Supplementary Fig.4) as mentioned above.

6) Figure 5 - ATP and GTP are shown, but what about other nucleotides? Evidence for methylthioadenosine seems weak/without context. Plots in this figure are not aligned. Heat maps would benefit from hierarchical clustering so that metabolites that trend similarly are positioned closely... as of now it appears to be alphabetical order or random order which do not add scientific meaning. Here and in general, many pertinent metabolites are missing - lactate, pyruvate, glucose, hexose phosphate, ribose phosphate, glyceraldehyde phosphate, succinate, fumarate.

Please see the complete metabolomics data set as mentioned above. Figure 5f and 5g show heatmaps for amino acids and fatty acid related metabolites, respectively. The hierarchical clustering is provided in the heatmap for all detected metabolites in this metabolomics analysis (Supplementary Fig.1, Supplementary Fig.5), and for amino acids (Supplementary Fig.3, Supplementary Fig.6 with azide inhibitor). The tissue lactate data is provided in Figure 7 and Supplementary Fig.2. The hexose phosphate data is shown in Supplementary Fig.2. For some reasons, the targeted metabolomics assay performed at UCLA does not reliably detect pyruvate and fumarate. Details of all quantification and statistics are provided in Source Data Supplementary Tables 4-8.

7) *Figure 5 - GSH and GSSG have the same trend. Have the authors explored de novo GSH synthesis (either enzymes or metabolites) or NADPH levels (glutathione reductase)?*

We have not analyzed enzymes involved in de novo GSH synthesis. However, following CNTF treatment we have detected elevated amino acids glutamate and glycine, which are involved in biosynthesis of GSH. In our metabolomics analysis NADPH is often not detected or detected with low intensities and may not be highly accurate in quantification.

8) *Figure 6 - hierarchical clustering needed in heat maps*

The rationale for Figure 6 organization is to allow easy assessment of metabolic pathways and biological processes. The RNA transcript heatmaps are presented in a way to allow the sequential steps of glycolytic pathway (Figure 6a) and TCA cycle (Figure 6b) to be visualized. Similarly, Figure 6d shows sequentially all nuclear genome-encoded respiratory chain complex gene transcripts. All RNA-sequencing data are provided in Source Data Supplementary Table 6.

9) *Figure 8 - "Ac-carnitine" is this meant to be "acylcarnitine" as in the legend, which is a very general name, or "acetylcarnitine" as in the C2 carnitine? Either way, more information is needed, like levels of palmitoylcarnitine. C2 carnitine does not need CPT to cross the mitochondrial membrane and is not a sufficient readout of carnitines at large.*

The shorthand Ac-carnitine in our data stands for acetyl-carnitine. We have now correctly indicated Acetyl-carnitine in Figure 5g, Figure 8c and 8i.

10) *typo in line 14*

Corrected.

11) *For a general readership like Nature Communications, the title needs to be flashier and CNTF should be defined.*

We changed the title to "Ciliary neurotrophic factor mediated neuroprotection involves profound alteration of retinal metabolism".

Reviewer #3

The manuscript NCOMMS-21-42240, "CNTF-mediated neuroprotection involves enhanced glycolysis and anabolic activities in degenerating retinas" is a comprehensive metabolomics study of the effects of ciliary neurotrophic factor (CNTF) in the rds mouse model of retinal degeneration. This work represents an extension of the authors' previous work on CNTF-mediated protection of photoreceptors (Rhee et al., 2013) and provides much-needed mechanistic insight into the biology underlying CNTF-mediated neuroprotection that is relevant to current human clinical trials. The authors are to be commended for their detailed investigation of the metabolic pathways underlying CNTF mediated protection in the rds mouse. They provide convincing evidence that CNTF has extensive effects on the metabolic state of the degenerating retina including suppression of mitochondrial respiratory chain function, promotion of aerobic glycolysis, and improvement in the retinal energy supply. Furthermore, they utilize robust metabolomics analyses to determine the contribution of glycolysis and mitochondrial respiration in healthy vs. degenerating retinas and suggest that CNTF-mediated protection is correlated with increases in aerobic glycolysis.

Comments:

1. CNTF was delivered via subretinal injection; it would be helpful to comment on the location and level of CNTF expression in retinal tissue to correlate with the histologic findings. I.e. are the images shown in Fig 1A-C at the site of the subretinal bleb, or were mitochondrial morphological changes seen diffusely in all retinal sections due to the secreted nature of CNTF? Were there any local inflammatory or histologic changes seen?

As noted by the reviewer, in this study we used the same lentiviral vector (LV-hCNTF) co-expressing a secreted form of recombinant human CNTF and EGFP (Rhee et al, PNAS, 2013; Reference 40). In our published paper (Reference 40), we presented data (fig.1) showing the structure of LV-hCNTF, the expression patterns of EGFP in RPE (not secreted) and the secreted CNTF protein permeated the entire retina. We also performed ELISA assay to determine the level of CNTF detected in the retina. In the same paper presented as supplemental data (fig S1), we demonstrated LV infection pattern that we observed routinely, including the initial subretinal injection bleb and the further spread of LV infection using EGFP as an indicator. More importantly, we showed (fig.5) that consistent with that the secreted CNTF indeed affects the entire mouse retina, LV-hCNTF treatment caused pan-retina responses indicated by the activation of CNTF downstream effectors STAT3 and ERK. Instead of showing these published data, for reviewers' convenience, below is the link to our previous publication and data:

www.pnas.org/lookup/suppl/doi:10.1073/pnas.1303604110/-/DCSupplemental.

The images of this current manuscript **Fig 1A-C** represent the typical rod cell mitochondrial morphologies under the different conditions (WT, rds, rds+ LV-IG, rds+LV-CNTF).

We did occasionally observe that in a minority (<5%) of LV-hCNTF injected eyes there were local ocular tissue inflammation as presented by cornea opacity. These eyes were excluded from the study.

2. How were the timepoints for LV-IG or LV-CNTF injection, harvesting, and various analyses determined, and do the metabolic effects of CNTF change if delivered in early vs. late-stage disease?

Given previous reports of decreased visual function after prolonged exposure to high levels of CNTF, were any additional time points assessed to determine if the metabolic profiles reported change over time? Were similar experiments conducted in more rapid murine models of retinal degeneration than the *rds* mouse and if so, did CNTF have a similar effect on the metabolic profile in these models?

The mouse model *rds/Prph2* (P216L) mutant we have used for this study undergoes progressive photoreceptor degeneration. If untreated by CNTF, by postnatal day 52 (P52), 80% of photoreceptors are lost as shown in our previous paper (fig.2, Rhee et al, PNAS, 2013; Reference 40, also see link above).

Fig. 2.

Thus, we have carefully chosen the time points to carry out different analyses in the current study. For most experiments, we started CNTF treatment at around postnatal day 25 (P25), when the *rds* retinas contain 85-90% of photoreceptors compared to the wild type control retina (see above). We have performed Seahorse respiration assays of whole retinas for CNTF treatment periods of P25-P35, P25-P42 (**Figure 3**), mitochondrial respiratory chain complex assays using purified mitochondria for P25-P49 (**Figure 4**). For these analyses, the CNTF treatment start time point is when 10-15% degeneration is discernable, and the end points are when non-treated *rds* retina still have 40-60% photoreceptors. For metabolomics analyses, we have assayed CNTF treated *rds* retinas from P25-P35 (**Figure 5**), P27-P45 and P27-P50 (**Figure 7**), P28-P53 (**Figure 8**). All results obtained from these assays consistently reveal that CNTF enhances glycolysis and anabolism, and increases retinal energy supply. We performed studies using these treatment periods to ensure that CNTF neuroprotection effects has taken place and the *rds* mutant retina still has some photoreceptor left.

Furthermore, our Western blot data has detected the increase and activation of LDHa in the *rds* retina up to P105 after CNTF treatment (**Figure 5e**). In the future, it would be interesting to compare metabolic profiles between older wild type retina (e.g. 6-12 months) and CNTF-treated *rds* retina to determine the long-term consequences of ageing and CNTF treatment. But an in-depth metabolic study of ageing retina and long term CNTF treatment is beyond the scope of the current report.

We have not performed metabolic assays in other rapid degeneration models. Previous studies have demonstrated CNTF's rescue effect on rod cells in the *rd1* mouse, which show a faster photoreceptor loss within 3 weeks postnatally (Azadi et al, 2007, Brain Res 1129:116). However, we have carried out genetic ablation of *SOCS3* gene, a direct negative regulator of CNTF downstream Jak-STAT signaling, in both the fast degeneration *rd10* mouse (Chang et al., 2007, Vision Res.,47:624) and *rds* models. Eliminating *SOCS3*-specifically on rod photoreceptor elevates CNTF signaling and leads to

photoreceptor rescues in both of the fast degeneration rd10 and the slow degeneration rds models. Moreover, we detected similar cellular signaling events with SOCS3 rod deletion and CNTF injection (Yang lab submitted). Interestingly, in the fast degenerating rd10 model, we also detected visual function preservation, which was not seen in the rds model. Thus, prolonged cytokine signaling required to achieve neuroprotection in rds retina may cause detrimental effects to visual function, possibly involving altered photoreceptor identity and remodeling of retinal network.

3. In Fig 2A, it appears that LV-IG, but not LV-CNTF improves the distribution isotropy. Conversely, in Fig 3B the basal OCR appears to be decreased in retinas treated with LV-CNTF and even lower in retinas treated with LV-IG. These counterintuitive findings should be clarified in the text.

This is a very insightful point. As discussed in our response to Reviewer#2 point 3), the apparent improvement of mitochondrial morphology and rescue of photoreceptors did not correlate with the oxygen consumption rate (OCR) (Figure 3b). The basal OCR was reduced following CNTF treatment and the ATP-linked OCR has returned to the wild type level (Figure 3d). This phenomenon was further confirmed with the respiratory chain complex assays using purified mitochondria (Figure 4), which showed suppressed complex I and IV activities. These results were initially surprising to us as we expected that CNTF might increase the respiratory chain activity as a consequence of improved mitochondrial morphology. However, the metabolomics analysis showed not only enhanced glycolysis, which is consistent with the ECAR increases (Figure 3), but also increase TCA cycle metabolites such as citrate and alpha-KG, which are key metabolites contributing to fatty acid and amino acid biosynthesis. Our current hypothesis is that CNTF signaling interferes with the respiratory chain complexes, leading to a partial uncoupling of the ETC and mitochondrial matrix activities as well as enhancing glycolysis and anabolic activity. We appreciate the insightful comments of the reviewer and have now discussed our hypothesis regarding these counterintuitive findings.

4. Clarity of Fig 6 could be improved by including transcriptome analyses of untreated rds mice, if available, in addition to WT, LV-IG, and LV-CNTF treated groups.

The reviewer is correct to point out the absence of untreated rds samples in the transcriptome data. In our transcriptome analysis, we found that PBS or LV-IG injected rds retinas show very similar transcriptome profile as untreated rds (Wang et al, 2020, Sci. Reports; Reference 44). Therefore, we used the control virus LV-IG injected samples to represent the rds condition, especially after a longer period post viral injection. Also, in our multiple batches of metabolomics assays, heatmaps repeated show similar metabolite profiles for rds and rds treated with the control LV-IG virus (see Figure S1 and Figure S2). Since our experiment involves subretinal injections, LV-IG injected retinas serves as proper controls for LV-CNTF injected retinas.

5. The authors' claims would be strengthened with evidence of visual function improvement in LV-CNTF treated rds animals at the time points assessed.

Despite that CNTF has been used in clinical trials, the field is still facing an enigma that CNTF-mediated neuroprotection often does not translate into visual function improvement. Most

published reports on CNTF in retinal neuroprotection have documented the disparity between cell preservation versus functional rescue. In fact, we do not consider nor claim that CNTF treatment in its current format is ideal. The objective of this study is to understand the underlying cellular mechanism of the potent CNTF neuronal survival effect in order to provide insight for the development of improved therapies.

We have shown that LV-CNTF transduction leads to effective preservation of photoreceptors in the rds mouse model, but did not result in improvement of visual function as measured by ERG in similar time windows (Rhee et al, PNAS, 2013; Reference 40). In our studies using the rd10 model, we have detected short-term visual function rescue by ablating the cytokine inhibitor SOCS3 in rod cells. However, this approach does not seem to result in visual function improvement in the slower degeneration rds model. Previous studies by other groups have shown visual function improvements in mouse and canine models (References 9, 11). The lack of visual function improvement despite robust neuroprotection could be contributed to CNTF signaling triggered transcriptome modulation as we have reported (Wang et al, 2020, Sci. Reports; Reference 44), but could also be attributed to different visual function assessments such as ERG versus optokinetic assays or brain imaging. Another factor to be considered is the differences among species and degeneration models. Most recently, CNTF trial for MacTel Type 2 has shown promising results using more sophisticated human visual function readouts than previous clinical trials. Since the underlying genetic deficiency for MacTel Type2 has been linked to amino acid biosynthetic pathway, our findings reported here provide an important mechanistic interpretation for the efficacy of CNTF treatment in the case of MacTel trial. Moreover, a multicenter CNTF trial for the major blinding disease glaucoma is ongoing, the findings reported in this study bridge an important gap in our understanding of CNTF's cellular function in the diseased retinas and provide important insight for designing more efficacious therapies.

Reviewer #4

Key Results: Ciliary neurotrophic factor (CNTF) is one of the most studied neurotrophic factors in the retina and exhibits potent neuroprotective activity in many models of retinal degeneration, however, mechanisms underlying its actions are incompletely understood. The authors of this paper investigated the influence of CNTF on retinal metabolism in a mouse model of retinitis pigmentosa. Employing a wide array of cutting-edge research technologies, they present a comprehensive evaluation of retinal metabolic status of the rds mutant and the action of CNTF. They show that CNTF improved photoreceptor mitochondrial morphology in the rds PRs. Metabolic flux respiration assays documented an attenuation of mitochondrial respiratory chain activity and increased glycolysis. Metabolomics analyses using glucose as a fuel revealed deficiencies in ATP, amino acids, and fatty acid biosynthetic intermediates, as well as a severe reduction in the major antioxidant glutathione in the rds retina. CNTF signaling exerted a strong influence on retinal metabolism in the rds retina, promoting glycolysis and stimulating anabolism. Transcriptome and metabolomics analyses revealed increased TCA cycle products, lipid biosynthetic pathway intermediates, nucleotides, and amino acids. Importantly, CNTF restored the key antioxidant glutathione to control values in the degenerating retina. This detailed and comprehensive study demonstrates that CNTF profoundly impacts the metabolic status of degenerating retinas by promoting aerobic glycolysis and anabolic metabolism revealing the fundamental cellular mechanism by which a neurotrophic factor promotes neuronal viability in disease conditions. These studies also provide

valuable mechanistic insight at the molecular and biochemical levels for the ongoing CNTF clinical trials for blinding diseases.

Validity: The data are presented clearly with detailed figures and illustrations. The data interpretation is consistent with the data presented.

Significance: The data presented in this manuscript make significant contributions to the literature from several perspectives. (1) Evidence that CNTF induces a profound change in retinal metabolism in retinal degeneration revealing a key cellular mechanism of cytoprotection. (2) The CNTF-induced shift from mitochondrial respiration to aerobic glycolysis is similar to the Warburg effect observed in cancer cells and pluripotent stem cells providing additional insight into the anabolic and cytoprotective nature of this metabolic shift. (3) Provided much needed mechanistic insights for ongoing clinical trials of CNTF for blinding diseases.

Data and methodology. An impressive aspect of this manuscript is the use of diverse and complementary methodology to address the questions. The data fit together nicely and the data obtained from the different methodological approaches support each other. Moreover, included in supplemental materials, the authors have included 5 videos of retinal histology, two metabolomic heat map figures, 6 tables containing all of the experimental data and a table of reagents.

Analytical approach: The analytical approach and statistics are valid and consistent with similar studies.

Suggested improvements: This manuscript is suitable for publication as it stands. However, given the effort expended in the mitochondrial morphological analysis (movies) and MitoLoc characterization, additional discussion of the functional implications of the differences in 3D morphology measured in the wild-type, rds and rds+CNTF would be valuable.

Clarity and context: The manuscript is clear, organized and well-written. The authors present their research in the context of previous studies conducted by them and by other investigators.

References: The manuscript references previous literature appropriately and comprehensively.

We appreciate the reviewer's evaluation of the quality of our study, its significance for understanding mechanism of CNTF-mediated neuroprotection, and the valuable insight for ongoing clinical trials aimed at treating retinal degeneration.

Regarding mitochondrial morphology versus function, the traditional view is that fragmental mitochondria have lower bioenergetic activities. Interestingly, we have detected diverse morphologies of mitochondria among different retinal cell types (Yang lab, unpublished), and even within the same neurons at different subcellular locations, e.g. rod mitochondria in the inner segment versus those at the ribbon synapse. We observed improved rod mitochondrial morphology by CNTF treatment but paradoxically attenuated respiratory chain function. We have now added statements in the discussion regarding the observed partial uncoupling between respiratory chain and mitochondrial matrix activities. Further research will be necessary to fully understand how CNTF signaling events in neuronal cells impact mitochondrial dynamics and overall cellular metabolism.

REVIEWER COMMENTS

Reviewer #1 (Remarks to the Author):

The authors have been very responsive in their revision and have well addressed all of the earlier comments. The revisions have strengthened the manuscript. This is an interesting study that will be of significance to the field and I recommend publication in Nature Communications.

Reviewer #2 (Remarks to the Author):

My concerns were sufficiently addressed (thank you) and i find the manuscript to be suitable for publication.

Reviewer #3 (Remarks to the Author):

The authors have improved the manuscript by including a more complete metabolomics data set and expanding the discussion of the paradoxical improvement in mitochondrial morphology but impaired respiratory chain function. The additional discussion of RP trial outcomes and relevance to the ongoing MacTel and glaucoma CNTF trials is also beneficial. Thank you for providing details of the lentiviral expression patterns from your previous publication (Rhee et al, 2013).

Given that this report studies retina-wide CNTF expression but only analyzes rod photoreceptor mitochondria it would be interesting to investigate mitochondrial function in other retinal cell types (ie RGCs), but this may be beyond the scope of the current report. Similarly, an assessment of the metabolic profiles over time or in a murine model of rapid degeneration would strengthen this manuscript, but may be beyond the scope of the project.

Response to Reviewers' Final Comments

Sept. 18, 2022

The entire comments of the reviewers are shown in an italic font. The authors' responses are shown below each reviewer's comments.

Reviewer #1 (Remarks to the Author):

The authors have been very responsive in their revision and have well addressed all of the earlier comments. The revisions have strengthened the manuscript. This is an interesting study that will be of significance to the field and I recommend publication in Nature Communications.

Reviewer #2 (Remarks to the Author):

My concerns were sufficiently addressed (thank you) and I find the manuscript to be suitable for publication.

We appreciate the insightful critiques of Reviewer #1 and Reviewer #2, which have helped us to improve the manuscript.

Reviewer #3 (Remarks to the Author):

The authors have improved the manuscript by including a more complete metabolomics data set and expanding the discussion of the paradoxical improvement in mitochondrial morphology but impaired respiratory chain function. The additional discussion of RP trial outcomes and relevance to the ongoing MacTel and glaucoma CNTF trials is also beneficial. Thank you for providing details of the lentiviral expression patterns from your previous publication (Rhee et al, 2013).

Given that this report studies retina-wide CNTF expression but only analyzes rod photoreceptor mitochondria it would be interesting to investigate mitochondrial function in other retinal cell types (ie RGCs), but this may be beyond the scope of the current report. Similarly, an assessment of the metabolic profiles over time or in a murine model of rapid degeneration would strengthen this manuscript, but may be beyond the scope of the project.

We thank Reviewer #3 for insightful and constructive comments. This study reveals the changes in metabolic status as a consequence of CNTF treatments in a photoreceptor degeneration model. We completely agree with Reviewer #3 that It is very important to explore the impact of CNTF on mitochondrial morphology and function as well as cellular metabolism of RGCs in glaucoma models. It will also be informative to examine the consequences of long-term CNTF treatments in various retinal degeneration models. These types of studies are currently being pursued to further advance mechanistic understanding of how neurotrophic factors promoting neuronal survival.